# Multi-view Masked Contrastive Representation Learning for Endoscopic Video Analysis

**Kai Hu**
Xiangtan University
kaihu@xtu.edu.cn

**Ye Xiao**
Xiangtan University
yxiao@smail.xtu.edu.cn

**Yuan Zhang**[*]
Xiangtan University
yuanz@xtu.edu.cn

**Xieping Gao**[*]
Hunan Normal University
xpgao@hunnu.edu.cn

## Abstract

Endoscopic video analysis can effectively assist clinicians in disease diagnosis and treatment, and has played an indispensable role in clinical medicine. Unlike regular videos, endoscopic video analysis presents unique challenges, including complex camera movements, uneven distribution of lesions, and concealment, and it typically relies on contrastive learning in self-supervised pretraining as its mainstream technique. However, representations obtained from contrastive learning enhance the discriminability of the model but often lack fine-grained information, which is suboptimal in the pixel-level prediction tasks. In this paper, we develop a **M**ulti-view **M**asked **C**ontrastive **R**epresentation **L**earning (M$^2$CRL) framework for endoscopic video pre-training. Specifically, we propose a multi-view masking strategy for addressing the challenges of endoscopic videos. We utilize the frame-aggregated attention guided tube mask to capture global-level spatiotemporal sensitive representation from the global views, while the random tube mask is employed to focus on local variations from the local views. Subsequently, we combine multi-view mask modeling with contrastive learning to obtain endoscopic video representations that possess fine-grained perception and holistic discriminative capabilities simultaneously. The proposed M$^2$CRL is pre-trained on 7 publicly available endoscopic video datasets and fine-tuned on 3 endoscopic video datasets for 3 downstream tasks. Notably, our M$^2$CRL significantly outperforms the current state-of-the-art self-supervised endoscopic pre-training methods, *e.g.*, Endo-FM (3.5% F1 for classification, 7.5% Dice for segmentation, and 2.2% F1 for detection) and other self-supervised methods, *e.g.*, VideoMAE V2 (4.6% F1 for classification, 0.4% Dice for segmentation, and 2.1% F1 for detection). [2]

## 1 Introduction

Video endoscopy is a crucial medical examination and diagnostic tool widely used for inspecting various tissues and structures (the digestive tract, respiratory tract, and abdominal cavity, *etc.*) [1]. In clinical practice, endoscopic video analysis usually relies on the experience and expertise of physicians, which is not only time-consuming and labor-intensive but also prone to subjective errors. Computer-aided medical analysis [2, 3, 4] can automatically and efficiently identify and classify

---

[*]Corresponding authors.

[2]Code is publicly available at: https://github.com/MLMIP/MMCRL.

38th Conference on Neural Information Processing Systems (NeurIPS 2024).

lesions, thereby assisting physicians in making more accurate diagnoses. In this paper, we focus on endoscopic videos with the aim of developing a robust pre-trained model for endoscopic video analysis to facilitate downstream tasks (*i.e.*, classification, segmentation, and detection).

Yann LeCun has mentioned "the revolution will not be supervised [5]" in multiple talks, emphasizing that the future development of artificial intelligence will increasingly rely on un-/self-supervised learning. Among them, self-supervised learning (SSL) aims to learn scalable visual representations from large amounts of unlabelled data for downstream tasks with limited annotated data. To learn meaningful representations for SSL, researchers crafted visual pretext tasks [6, 7, 8, 9, 10, 11, 12, 13, 14, 15, 16], which are summarized into two main categories: contrastive and generative [17]. Contrastive methods, also known as discriminative methods, employ a straightforward discriminative idea that pulling closer representations from the same image and pushing away different images, *i.e.*, contrastive learning (CL) [6, 7, 8]. By utilizing image-level prediction with global features, CL can naturally endow pre-trained models with strong instance discriminability, which has been proven to be effective in classification tasks. However, CL also presents the challenge that downstream dense prediction tasks, such as segmentation and detection, are not fully considered.

Generative methods aim to reconstruct the input data itself by encoding the data into features and then decoding it, including AE [18], VAE [19], GAN [20], *etc*. Recently, masked image modeling (MIM) [13, 14, 16] has demonstrated the strong potential in self-supervised learning. MIM masks a substantial portion of image patches during training and utilizes an autoencoder [18] to reconstruct the original signal of the image, which, unlike CL, can enhance the ability to capture the pixel-level information. Following the success of MIM, some works have tried to extend this new pre-training paradigm to the video domain for self-supervised video pre-training [21, 22, 23]. Actually, mask techniques [24, 25, 26] are crucial for the success of mask modeling. Endoscopic videos, in particular, have higher dimensions and redundancy compared to static images. They also exhibit unstable inter-frame variations due to the manual manipulation by the doctor. Additionally, lesions in endoscopic videos often have low contrast and appear in obscured regions or with subtle variations. Therefore, simply applying random mask not only requires extensive pre-training time but also easily leads to poor performance.

To address the aforementioned issues, in this paper, we develop a Multi-view Masked Contrastive Representation Learning framework named $M^2$CRL. **First**, considering the characteristics of inter-frame instability and small inter-class differences in endoscopic videos, we propose a multi-view masking strategy. Specifically, we introduce a frame-aggregated attention guided tube masking strategy for the global views, which aggregates features from multiple frames to capture global spatiotemporal information. Simultaneously, a random tube masking strategy is employed from the local views, enabling the model to focus on local features. **Second**, to address the inadequacy of capturing pixel-level details in contrastive learning, we integrate multi-view masked modeling into contrastive approach, which not only encourages the model to learn discriminative representations but also forces it to capture more refined pixel-level features. Extensive experiments have verified that our $M^2$CRL significantly enhances the quality of endoscopic video representation learning and exhibits excellent generalization capabilities in multiple downstream tasks (*i.e.*, classification, segmentation and detection). Overall, our contributions are summarized as follows:

- We propose a novel multi-view masking strategy aimed at enhancing the capture of fine-grained representations in endoscopic videos by performing mask modeling on both global and local views. This strategy involves utilizing the frame-aggregated attention guided tube mask to capture global-level spatiotemporal contextual relationships from the global views, while employing the random tube mask to focus on local variations from the local views.

- We propose a multi-view masked contrastive representation learning framework that combines multi-view mask modeling with contrastive method to train endoscopic videos, which effectively addresses the limitation of contrastive method in capturing dense pixel dependencies by predicting the intensity of each pixel within masked patches.

- We conduct extensive experiments on 10 endoscopic video datasets to evaluate the performance of $M^2$CRL in comparison to other methods. $M^2$CRL achieves 94.2% top-1 accuracy on PolypDiag [27], 81.4% on CVC-12k [28], and 86.3% on KUMC [29], outperforming the state-of-the-art methods, *i.e.*, Endo-FM [30] by +3.5%, +7.5%, and +2.2%, and VideoMAE V2 by +4.6%, +0.4%, and +2.1%, respectively.

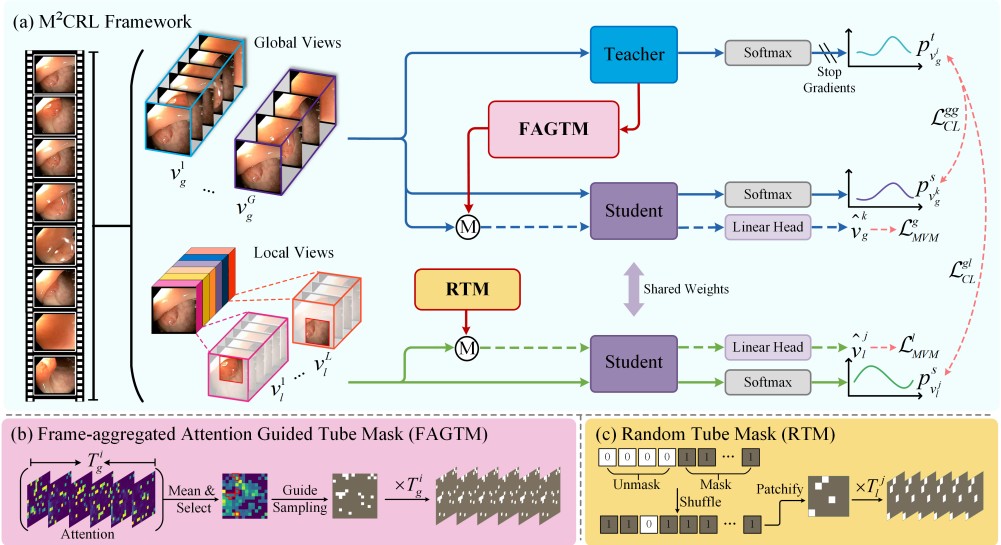

Figure 1: The pipeline of the proposed M$^2$CRL. For the generated global and local views with different frame rates and spatial sizes, we adopt two different mask strategies: the frame-aggregated attention guided tube mask and the random tube mask. These strategies are integrated with mask reconstruction and contrastive method, enabling the model to simultaneously learn both the pixel-level and discriminative features of the video.

## 2 Related Work

### 2.1 Self-supervised video representation learning

Self-supervised learning is a machine learning technique that mitigates the reliance on manual labeling by exploiting the inherent structure or properties of the data itself, and its common goal in computer vision is to learn scalable and generalisable visual representations. In SSL, the key lies in designing an appropriate pretext task, which involves generating suitable supervision using only visual signals. Pretext tasks in images [9, 10, 11, 12] have been widely explore, and motivated by images, there has been great interest in exploring SSL pretext tasks for videos. Some early works [31, 32, 33] focused on extending image SSL methods for video, however video has a specific temporal dimension. The temporal information can be leveraged for representation learning, including future prediction [34, 35, 36, 37, 38, 39, 40], temporal ordering [41, 42, 43, 44, 45], object motion [46, 47, 48, 49], and temporal consistency [50, 51]. In the last few years, contrastive learning [52, 53, 54, 55, 56, 57] has made great progress in SSL. However, SSL based on contrastive methods typically focuses on discriminative features of holistic information, while lacking the ability to focus on fine-grained features.

### 2.2 Masked visual modeling

Masked visual modeling is proven to be a simple and effective self-supervised learning paradigm [13, 14, 15, 16, 58, 21, 22] through a straightforward pipeline based on masking and reconstruction. This paradigm quickly expanded to the video domain, *e.g.*, VideoMAE [22] and ST-MAE [21], which extend image MAE to video. As with images, the choice of masking strategy affects self-supervised representation learning [24, 25, 26, 59]. VideoMAE [22] employs random tube mask to model the same locations across frames to ensure spatial discontinuity while maintain temporal continuity. In contrast, ST-MAE [21] generates a random mask for each frame independently, which are discontinuous in both space and time. Nevertheless, mask video modeling using randomly masked tokens to reconstruct is inefficient because the tokens embedded in a video frame are not equally important. Several studies [60, 61, 62, 63] have proposed various motion-based video mask strategies. However, our focus is on endoscopic videos, which lack a strong motion counterpart compare to the above works.

## 2.3 Self-supervised learning for endoscopy

In recent years, SSL has received increasing attention in the field of medical analysis, including endoscopy. Endo-FM [30] employs contrastive method to minimize the disparity in feature representations between different spatiotemporal views of the same video. Intrator [64] has proposed the use of contrastive learning to adapt video inputs for appearance-based object tracking. Hirsch [65] applies the SSL framework masked siamese networks (MSNs) to analyze endoscopic videos. While MSNs use a masking concept, it primarily serves as a data augmentation technique, with its essence still rooted in contrastive learning. Currently, most self-supervised pre-training for endoscopic videos relies on contrastive methods. While these methods have shown promise for endoscopic video pre-training, relying solely on them may not fully capture the fine feature expressions of endoscopic videos.

# 3 Method

To address the limitations of contrastive methods in fine-grained information perception in SSL for endoscopic videos, we propose a Multi-view Masked Contrastive Representation Learning (M$^2$CRL) framework for endoscopic video pre-training, as shown in Fig. 1. Here, we first review masked prediction in Section 3.1. Then, we present our proposed multi-view masking strategy in Section 3.2 and introduce our M$^2$CRL framework in Section 3.3.

## 3.1 Preliminary

Masked prediction is a prevalent representation learning technique in natural language processing (NLP) [66, 67, 68, 69], and many researchers have explore its application to images [14, 16, 70, 71, 72] and videos [21, 22, 23]. MIM endeavors to learn image representations by solving a regression problem, where the model is tasked with predicting the pixel values in randomly masked patches of the image. Specifically, an image $x \in \mathbb{R}^{H \times W \times C}$ is reshaped into $N = HW/P^2$ flattened patches as $\{x_i\}_{i=1}^N$, where $(H, W)$ is the resolution of image, $C$ is the number of channels and $P$ is the patch size. Each patch is represented with token embedding. MIM constructs a random mask $m \in \{0, 1\}^N$ to indicate the masked tokens that correspond to $m_i = 1$. In MAE [14], only the visible tokens $\{x_i | m_i = 0\}_{i=1}^N$ are fed into the vision transformer to obtain the latent feature, and then the decoder uses the latent feature and the masked tokens as inputs to predict $\hat{y}$. In SimMIM [16], visible and invisible tokens are fed into the encoder. The prediction loss is calculated as the loss between the normalized masked tokens and the reconstructed ones in the pixel space by:

$$\mathcal{L} = \frac{1}{N_m} \sum \|\hat{y}_m - x_m\|_p \tag{1}$$

where $N_m$ is the number of masked tokens, $x_m$ is the masked token, $p$ is norm and its value is 1 or 2.

## 3.2 Multi-view masking strategy

Masked video modeling (MVM) [21, 22, 23] employs random mask strategies (*i.e.*, random, spatial-only, temporal-only) to capture meaningful representations from pre-training videos. Although these strategies are effective for general video datasets with well-curated and stable distributions, they do not account for the unique characteristics of medical data. We have summarized two key characteristics of endoscopic videos: (1) Instability of inter-frame variations is a prominent feature of endoscopic videos. These variations are driven by camera movement, instrument manipulation, and the uneven distribution of lesion areas, *e.g.*, variations can range from drastic to minor, as the camera navigates from the intestinal wall to specific lesion sites. (2) Endoscopic video exhibits characteristics of small inter-class differences. The lesion tissues typically resemble surrounding the normal tissues in color, texture, or shape, which complicates the model's ability to accurately identify the lesion area. Therefore, we propose a multi-view masking strategy by considering the above two points, the details are as follows.

### 3.2.1 Frame-aggregated Attention Guided Tube Mask

To address the challenge of instability between video frames, we propose a frame-aggregated attention guided tube masking strategy. We aggregate the attention of all frames of the video along the frame

dimension to generate a frame-aggregated attention map, which then dynamically guides the masking process. This way can capture the overall scene information in a video sequence from the global spatiotemporal information and ignores irrelevant spatiotemporal noise to some extent.

**Semantic information extraction**  Our architecture consists of the teacher network $f_t$ and the student network $f_s$. Our network employs a self-attention mechanism known as divided space-time attention mechanism [73], which enhances the learning ability of the network while reducing the computational complexity. Specifically, we take an endoscopic video, from which we sample the global views $\{v_g^i \in \mathbb{R}^{T_g^i \times 3 \times H_g \times W_g}\}_{i=1}^G$, where $T_g$ is the number of frames in the sampled view. Each frame is then divided into $N = H_g W_g / P^2$ patches, which are mapped into patch tokens and fed into the transformer blocks of the teacher network. Thus, each encoder block processes $N$ patch (spatial) and $T_g$ temporal tokens. The network includes a learnable class token, $[cls]$, which represents the global features learned by the network along spatial and temporal dimensions. Given the intermediate token $e^u \in \mathbb{R}^{(N+1) \times D}$ from block $u$, the token in the next block is computed as follows:

$$\begin{cases} e_{time}^{u+1} = MSA_{time}\left(LN\left(e^u\right)\right) + e^u, \\ e_{space}^{u+1} = MSA_{space}\left(LN\left(e_{time}^{u+1}\right)\right) + e_{time}^{u+1}, \\ e^{u+1} = MLP\left(LN\left(e_{space}^{u+1}\right)\right) + e_{space}^{u+1} \end{cases} \tag{2}$$

where $MSA$, $LN$ and $MLP$ denote the multi-head self-attention, layer normalization, and multi-layer perceptron, respectively. Each block utilizes $MSA$ layer to project and divide $e$ into $n_h$ parts. Each part contains the query $Q_r$, key $K_r$ and value $V_r$ for $r = 1, 2, ..., n_h$, where $n_h$ denotes the number of heads. We can get the attention map of the last layer of blocks by calculating the correlation between the query embedding of class token $Q^{cls}$ and key embeddings of all other patches $K$. It is averaged for all heads as follows:

$$A = \frac{1}{n_h} \sum_{r=1}^{n_h} Softmax(Q_r^{cls} \frac{K_r}{\sqrt{D_h}}) \tag{3}$$

where $D_h = D/n_h$, and $A$ is $T_g$ spatial attention maps. Although single-frame spatial attention integrates temporal information, it still only considers the spatial information of the current frame, neglecting the global spatiotemporal dependencies in the video sequence. Thus, we aggregate the attention of $T_g$ frames to the mean value to obtain a simplified and holistic attention distribution by:

$$A_{agg} = \frac{1}{T_g} \sum_{t=1}^{T_g} A_t \tag{4}$$

This attention mechanism is capable of obtaining an approximation of the critical region of the video that are being attended to from both the temporal and spatial dimensions, while reducing the impact of excessive variations in individual frames or regions. We will further exploit this attention dynamic to guide the generation of tube masking to help the model perform the reconstruction task more appropriately.

**Visible tokens sampling and masking**  The traditional random masking strategy treats critical and non-critical areas of the video equally in each iteration, which may lead to excessive masking of key video regions at a high masking ratio, thereby affecting the learning ability of the model. Therefore, we utilize a frame-aggregated attention mechanism to guide the generation of tube mask. By sampling some reasonably visible tokens from the model-focused areas and masking the rest, it allows our method to efficiently perform the reconstruction task even at a high masking ratio, while improving pre-training efficiency. Specifically, we begin by ranking tokens in descending order of attention scores and subsequently select a proportion of high-attention tokens based on the threshold $\gamma$. We randomly sample visible tokens to form a binary mask from selected tokens of attention, as depicted in Fig. 1(b). The number of sampled visible tokens $N_v$ is determined by the predefined mask ratio $\rho \in (0, 1)$. Followed by SimMIM [16], we fill the mask tokens with learnable mask embeddings, which are subsequently fed together into the student network for feature mapping and finally into the prediction head for reconstruction.

In comparison to random sampling, which is inefficient in token allocation, our method selects visible tokens based on the global spatiotemporal information of the video sequence. This approach

Table 1: Comparison with other latest SOTA methods on 3 downstream tasks. We report F1 score (%) for PolypDiag, Dice (%) for CVC-12k, and F1 score (%) for KUMC, respectively.

| Method | Venue | Year | Pretrain Time(h) | PolypDiag (Classificaton) | CVC-12k (Segmentation) | KUMC (Detection) |
|---|---|---|---|---|---|---|
| Scratch (Rand.init.) | - | - | N/A | 83.5 ± 1.3 | 53.2 ± 3.2 | 73.5 ± 4.3 |
| TimeSformer [73] | ICML | 2021 | 104.0 | 84.2 ± 0.8 | 56.3 ± 1.5 | 75.8 ± 2.1 |
| CORP [74] | ICCV | 2021 | 65.4 | 87.1 ± 0.6 | 68.4 ± 1.1 | 78.2 ± 1.4 |
| FAME [75] | CVPR | 2022 | 48.9 | 85.4 ± 0.8 | 67.2 ± 1.3 | 76.9 ± 1.2 |
| ProViCo [76] | CVPR | 2022 | 71.2 | 86.9 ± 0.5 | 69.0 ± 1.5 | 78.6 ± 1.7 |
| VCL [77] | ECCV | 2022 | 74.9 | 87.6 ± 0.6 | 69.1 ± 1.2 | 78.1 ± 1.9 |
| ST-Adapter [78] | NeurIPS | 2022 | 8.1 | 84.8 ± 0.7 | 64.3 ± 1.9 | 74.9 ± 2.9 |
| VideoMAE [22] | NeurIPS | 2022 | 25.3 | 91.4 ± 0.8 | 80.9 ± 1.0 | 82.8 ± 1.9 |
| Endo-FM [30] | MICCAI | 2023 | 20.4 | 90.7 ± 0.4 | 73.9 ± 1.2 | 84.1 ± 1.3 |
| DropMAE [79] | CVPR | 2023 | 37.9 | 88.2 ± 0.8 | 80.9 ± 0.3 | 81.7 ± 2.6 |
| VideoMAE V2 [23] | CVPR | 2023 | 17.3 | 89.6 ± 1.4 | 81.0 ± 0.4 | 84.2 ± 1.0 |
| M$^2$CRL | Ours | - | 24.3 | **94.2 ± 0.7** | **81.4 ± 0.8** | **86.3 ± 0.8** |

minimizes redundant background area sampling, as these backgrounds have minimal impact on the significance of mask reconstruction. Furthermore, the attention-guided mask generation is dynamic and allows for adjustments during model training. This adaptability enables the model to continuously optimize its focus, adapting to complex or changing data characteristics.

### 3.2.2 Random Tube Mask

Global view mask modeling is primarily employed for a comprehensive understanding and spatiotemporal contextual awareness of the entire endoscopic video frame, which mitigates the effects of inter-frame variability instability by aggregating frame attention. However, due to the low contrast between lesions and normal tissues in endoscopic videos, global views may struggle to accurately capture local details. Hence, we apply random tube mask reconstruction on local views to learn more granular detail information, as shown in Fig. 1(c). Specifically, we obtain the local views $\{v_l^j = \mathbb{R}^{T_l^i \times 3 \times H_l \times W_l}\}_{j=1}^L$ ($T_l < T_g$) by random cropping and uniform sampling at different frame rates. In local views, we also implement a high masking ratio $\rho = 90\%$ to reduce information leakage during the mask modeling process. By local view mask modeling with different frame rates and spatial cropping, it is possible to make the model proficient in capturing variations in time scale and spatial detail. Moreover, local view mask modeling focuses on specific regions in the video without interference from the global background, enables more targeted learning. This allows the model to finely capture local information within the video, enhancing its ability to recognize subtle differences between abnormalities and normal tissues.

### 3.3 Multi-view Masked Contrastive Representation Learning

The pipeline of our proposed M$^2$CRL is shown in Fig. 1(a), which introduces a multi-view masking strategy and combines multi-view mask modeling with contrastive learning to learn representations that possess fine-grained and discriminative capabilities simultaneously.

In DINO [55], self-distillation is proposed not from the posterior distribution but by a teacher-student scheme that extracts knowledge from the model's own past iterations. This self-distillation method of self-supervision is also considered a form of contrastive learning [80]. The contrastive learning part of our M$^2$CRL follows Endo-FM [30], which also employs self-distillation method to achieve representation learning. Given an endoscopic video, two types of views are created under random data augmentation ($G$ global views $\{v_g^i\}_{i=1}^G$ and $L$ local views $\{v_l^j\}_{j=1}^L$). The model is encoded by two encoders, a teacher network $f_t$ and a student network $f_s$, which are respectively parameterized by $\theta_t$ and $\theta_s$. It should be noted that the two student networks depicted in Fig. 1(a) actually represent

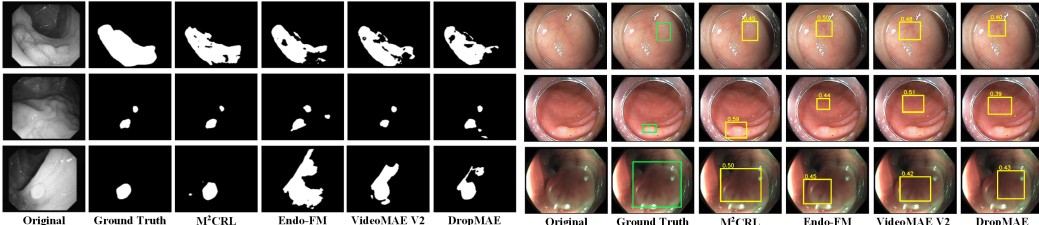

| Original | Ground Truth | M²CRL | Endo-FM | VideoMAE V2 | DropMAE | Original | Ground Truth | M²CRL | Endo-FM | VideoMAE V2 | DropMAE |

Figure 2: Qualitative results of segmentation and detection tasks. The segmentation results on the left are from the CVC-12k dataset, while the detection results on the right are from the KUMC dataset.

a single student network. We illustrate two student networks in the figure to more clearly convey the data flow. During pre-training, the global views are input into both the teacher and student networks, while the local views are only input into the student network. The network output $f$ is normalized by a softmax function with a temperature $\tau$ to obtain the probability distribution $p = \text{softmax}(f / \tau)$. Subsequently, the cross-entropy loss function is used to compute the losses between the teacher's global views and the student's global views, as well as between the teacher's global views and the student's local views. The specific loss functions are as follows:

$$
\begin{aligned}
\mathcal{L}_{CL}^{gg} &= \sum_{i=1}^{G} \sum_{\substack{k=1 \\ k \neq i}}^{G} -p_{v_g^i}^t \log p_{v_g^k}^s, \\
\mathcal{L}_{CL}^{gl} &= \sum_{i=1}^{G} \sum_{j=1}^{L} -p_{v_g^i}^t \log p_{v_l^j}^s
\end{aligned}
\tag{5}
$$

Contrastive learning utilizes class tokens [55] from global and local views to calculate matching losses, which offers strong capabilities for overall discriminative representation. However, this method overlooks the dense pixel dependencies that are crucial for dense prediction tasks like segmentation and detection. Therefore, we integrate the multi-view mask reconstruction pre-training task into the contrastive learning. In the multi-view mask modeling task, the video clips processed through masking are fed into the encoder to be mapped into the feature space. Subsequently, the prediction head utilizes the context of unmasked patches to regress the dense pixel intensities within masked patches. The losses for global and local view reconstruction are as follows:

$$
\begin{aligned}
\mathcal{L}_{MVM}^{g} &= \frac{1}{N_m^g} \sum_{i=1}^{G} \left\| \hat{v}_g^i - v_g^i \right\|, \\
\mathcal{L}_{MVM}^{l} &= \frac{1}{N_m^l} \sum_{j=1}^{L} \left\| \hat{v}_l^j - v_l^j \right\|
\end{aligned}
\tag{6}
$$

M²CRL exploits both contrastive loss and reconstruction loss in optimization, and the total loss is $\mathcal{L}_{total} = \mathcal{L}_{CL}^{gg} + \mathcal{L}_{CL}^{gl} + \mathcal{L}_{MVM}^{g} + \mathcal{L}_{MVM}^{l}$. The student network updates the student parameters $\theta_s$ by minimizing $\mathcal{L}_{total}$, and the teacher network $\theta_t$ is updated using the exponential moving average (EMA) of the student weights, and $\theta_t \leftarrow \lambda \theta_t + (1 - \lambda) \theta_s$. Here $\lambda$ denotes the momentum coefficient. By combining contrastive learning and masked video modeling, the model is encouraged to learn the holistic discriminative representation and detailed pixel information, effectively improving the learning ability of the model in complex visual data of endoscopic videos.

## 4 Experiments

### 4.1 Datasets and Experimental Settings

We conduct experiments on 10 publicly available endoscopic video datasets: Colonoscopic [81], SUN-SEG [82], LDPolypVideo [83], Hyper-Kvasir [84], Kvasir-Capsule [85], CholecTriplet [86], Renji-Hospital [30], PolypDiag [27], CVC-12k [28], and KUMC [29]. These datasets have a total of 33,231 videos with approximately 5,500,000 frames, covering 3 types of endoscopy examination protocols and 10+ different diseases. The videos are processed into 30fps clips with an average duration of 5 seconds. The first 7 datasets are used for pre-training and we sample $G = 2$ global views

and $L = 8$ local views, where $T_g \in [8, 16]$, $T_l \in [2, 4, 8, 16]$ and the spatial size set to $224 \times 224$ and $96 \times 96$, respectively. We take ViT-B/16 [87] as the backbone and perform 30 epochs of pre-training. In downstream tasks, we perform classification task on PolypDiag, segmentation task on CVC-12k, and detection task on KUMC, respectively. Our implementation is based on Endo-FM [30], and more experimental details can be found in § A.

## 4.2 Comparison with Prior Work

We compare our method with the recent state-of-the-art (SOTA) endoscopic video pre-training model, Endo-FM [30], which is the first pre-training model on a large scale across various endoscopic videos. The results of other methods are taken from the comparative method of Endo-FM, including TimeSformer [73], CORP [74], FAME [75], ProViCo [76], VCL [77], and ST-Adapter [78]. Additionally, we also compare the latest video self-supervised methods: VideoMAE [22], VideoMAE V2 [23] and DropMAE [79]. For a fair comparison, all methods are pretrained on the same union of 7 datasets as our $M^2$CRL. All experimental settings are referred to those documented in the original papers or in the released code.

**Quantitative evaluation**    We observe that our method outperforms existing state-of-the-art methods, as shown in Table 1. Particularly, compared to Endo-FM [30], our $M^2$CRL achieves improvements of 3.5% F1 in classification (PolypDiag), 7.5% Dice in segmentation (CVC-12k), and 2.2% F1 in detection (KUMC) tasks. These improvements are attributed to our pre-training approach, which integrates multi-view mask modeling with contrastive method, substantially enhancing the model's ability for representation learning. Clearly, compared to other methods (TimeSformer [73], CORP [74], FAME [75], *etc.*), $M^2$CRL achieves considerable advantages across three downstream tasks. Although the performance gains on extremely dense task (*i.e.*, segmentation) is modest compared to the latest video self-supervised methods (*i.e.*, VideoMAE [22], VideoMAE V2 [23], DropMAE [79]), our $M^2$CRL makes great progress on classification task and reaches 94.2%. This shows that our $M^2$CRL not only focuses on pixel details but also enhances discriminative capabilities.

**Qualitative evaluation**    We visualize segmentation and detection results in Fig. 2. Compared to other methods, our $M^2$CRL demonstrates superior visual results in segmenting both large and small polyp regions ($1^{st}$ and $2^{nd}$ rows on the left in Fig. 2). Despite potential issues such as blurry boundaries or lens glare caused by camera movement, $M^2$CRL is still capable of accurately segmenting the target regions ($3^{rd}$ row on the left in Fig. 2). We attribute these results to our multi-view mask modeling, which encourages the model to learn more precise detail information from videos. Similarly, $M^2$CRL also exhibits good performance in detection tasks, especially in detecting small target regions ($2^{nd}$ row on the right in Fig. 2). Although there are some differences between the predictions of our method and the ground truth, we achieve a high degree of overlap with the ground truth. This demonstrates that our $M^2$CRL significantly improves the pre-training capability for endoscopic videos. See § D for more segmentation and detection visual comparison.

## 4.3 Ablation Studies

**Multi-view mask**    Table 2 illustrates the impact of single-view mask and multi-view mask. All experiments use the same masking ratio. For the single-view, different masking strategies are employed for global and local views, respectively. The random tube mask [22] randomly samples masked tokens in the 2D spatial domain and then extends these tokens along the temporal axis. The random mask [21] randomly masks tokens in the spatiotemporal domain. Our approach samples visible tokens by leveraging frame-aggregated attention from global views, resulting in better results than above two strategies, particularly achieving 80.6% in segmentation task. Similarly, the random tube mask demonstrates some superiority on local views. However, from Table 2, it can be observed that the performance of solely conducting mask modeling on the single-view is inferior to that on multi-views. In multi-view mask, it is evident that employing the frame-aggregated attention guided tube mask on global views and the random tube mask on local views results in significant performance gains. These two complementary mask methods work together to learn richer details of video features.

**Hyper-parameters of the FAGTM**    We conduct an experiment to investigate the impact of the hyperparameters of the frame-aggregated attention guided tube mask (FAGTM) for global views,

Table 2: **Multi-view mask**. We compare multiple different masking strategies on different views. FAGTM = Frame-aggregated Attention Guided Tube Mask. RTM = Random Tube Mask.

| views | mask strategies | | cla. | seg. | det. |
|---|---|---|---|---|---|
| | gloabl | local | | | |
| single-view | random | - | 90.2 ± 1.5 | 78.6 ± 1.6 | 83.8 ± 1.9 |
| | RTM | - | 93.0 ± 0.8 | 77.5 ± 3.1 | 84.0 ± 1.0 |
| | FAGTM | - | 92.7 ± 0.4 | 80.6 ± 0.5 | 84.4 ± 1.4 |
| | - | random | 91.1 ± 0.7 | 76.1 ± 1.5 | 83.7 ± 0.7 |
| | - | RTM | 91.1 ± 0.5 | 77.5 ± 0.6 | 85.0 ± 0.4 |
| multi-view | random | random | 91.3 ± 0.3 | 77.7 ± 0.4 | 84.9 ± 1.0 |
| | RTM | RTM | 93.2 ± 0.4 | 80.2 ± 0.9 | 85.2 ± 1.3 |
| | FAGTM | RTM | **94.2 ± 0.7** | **81.4 ± 0.8** | **86.3 ± 0.8** |

Table 3: **Hyper-parameters** of the FAGTM.

| $\gamma$ | cla. | seg. | det. |
|---|---|---|---|
| 0.5 | 91.6 ± 0.5 | 80.7 ± 0.7 | 84.8 ± 0.4 |
| 0.6 | **94.2 ± 0.7** | **81.4 ± 0.8** | **86.3 ± 0.8** |
| 0.7 | 93.7 ± 1.1 | 81.0 ± 0.1 | 85.9 ± 2.6 |
| 0.8 | 92.9 ± 0.7 | 79.0 ± 0.3 | 85.4 ± 0.8 |

Table 4: **The teacher's block** of the FAGTM.

| blocks | cla. | seg. | det. |
|---|---|---|---|
| 4 | 91.8 ± 0.7 | 76.6 ± 1.5 | 83.6 ± 1.1 |
| 8 | 92.5 ± 0.4 | 79.7 ± 2.2 | 84.8 ± 1.0 |
| 10 | 93.9 ± 1.0 | 80.6 ± 0.9 | 85.9 ± 1.7 |
| 12 | **94.2 ± 0.7** | **81.4 ± 0.8** | **86.3 ± 0.8** |

the results are shown in Table 3. We perform experiments by sorting each patch of the obtained frame-aggregated attention maps in descending order and selecting the top $\gamma$ proportion of patches as candidate mask patches. Subsequently, visible tokens are random sampled in candidate mask patches. From Table 3, we can observe that the model performs best when $\gamma$ is set to 0.6. A lower value indicates selecting visible patches from smaller high-attention regions, leading to excessive attention on non-critical areas during reconstruction, contradicting the setup of the self-supervised pre-text task. On the other hand, higher values of $\gamma$ adversely affecting the learning efficacy of the model.

**The teacher's block used for the FAGTM** In our study, we use the last layer block of the teacher ViT-B for the FAGTM. The higher layer block incorporates lower-level features and object-level semantic information, offering comprehensive and abstract features that are essential for the model. Thus, it effectively guides the student network in masking. Table 4 shows that it is most beneficial for the FAGTM to utilize the last layer block of the teacher network.

**Masking ratio** The impact of different masking ratios is illustrated in Table 5. It can be observed that there is an improvement in results across three downstream tasks when the masking ratio increases from 75% to 90%. Due to the reundancy in videos, it shows that the 75% masking ratio utilized in ImageMAE is not suitable for videos. When the masking ratio in videos reaches 90%, the task becomes challenging due to the limited number of patches available for learning temporal correspondence, thus enhancing the learning capacity of the model. This observation is also validated in VideoMAE [22]. Compared to the optimal masking ratio, a higher masking ratio increases the difficulty of pre-training, hindering the model's ability to learn effective representations. Although our ablation experiments have shown that a masking ratio of 95% can achieve comparable performance, its effectiveness in three downstream tasks is lower than that of the 90% masking ratio. This suggests that at a masking ratio of 95%, the model is placed in a relatively unfavorable learning situation, resulting in suboptimal results.

**Analysis of components** As see from the first row in Table 6, although the performance of the contrastive learning framework on classification tasks is acceptable, its performance on pixel-level tasks, especially the segmentation task, is not very good. Similarly, within the single mask modeling

Table 5: **Masking ratio.** We choose a masking ratio of 90% for the FAGTM and RTM.

| FAGTM (global) | RTM (local) | cla. | seg. | det. |
|---|---|---|---|---|
| | 95% | 94.0 ± 0.3 | 81.3 ± 0.4 | 85.1 ± 1.1 |
| 95% | 90% | 93.8 ± 0.9 | 80.7 ± 0.7 | 86.2 ± 1.3 |
| | 85% | 93.2 ± 0.7 | 78.5 ± 0.6 | 84.9 ± 2.3 |
| | 75% | 92.6 ± 0.4 | 77.4 ± 1.7 | 85.2 ± 2.1 |
| | 95% | 93.8 ± 1.4 | 80.5 ± 0.7 | 85.8 ± 0.9 |
| 90% | 90% | **94.2 ± 0.7** | **81.4 ± 0.8** | **86.3 ± 0.8** |
| | 85% | 93.8 ± 0.8 | 81.4 ± 1.7 | 85.6 ± 2.2 |
| | 75% | 93.2 ± 0.9 | 78.5 ± 1.9 | 84.8 ± 1.3 |
| | 95% | 93.2 ± 0.8 | 79.9 ± 0.4 | 83.1 ± 1.5 |
| 85% | 90% | 93.8 ± 0.2 | 81.2 ± 0.2 | 83.8 ± 0.9 |
| | 85% | 94.0 ± 0.4 | 80.5 ± 1.0 | 85.1 ± 1.8 |
| | 75% | 92.5 ± 1.2 | 79.6 ± 0.7 | 83.8 ± 2.5 |
| | 95% | 91.7 ± 1.3 | 76.8 ± 1.8 | 84.2 ± 2.1 |
| 75% | 90% | 91.3 ± 0.3 | 79.0 ± 2.2 | 84.0 ± 1.3 |
| | 85% | 91.8 ± 0.2 | 77.5 ± 1.9 | 83.8 ± 0.4 |
| | 75% | 91.2 ± 0.7 | 74.6 ± 1.4 | 85.0 ± 0.8 |

self-supervised framework, there is an improvement in performance for the pixel-level task (*e.g.*, from 73.9% to 80.5% for segmentation), while there is a significant drop in performance for the discriminative task (*e.g.*, from 90.7% to 85.7% for classification). These interesting phenomena once again highlights the strong instance discrimination ability of contrastive learning and the robust ability of mask modeling to acquire local pixel-level information. Our approach integrates both methods and demonstrates consistently strong performance across both structural and pixel-level tasks.

Table 6: **Components analysis**. Our proposed $M^2$CRL combines masked video modeling with contrastive learning and has the best performance.

| contrastive learning | masked video modeling | cla. | seg. | det. |
|---|---|---|---|---|
| ✓ | | 90.7 ± 0.4 | 73.9 ± 1.2 | 84.1 ± 1.3 |
| | ✓ | 85.7 ± 0.4 | 80.5 ± 1.2 | 83.5 ± 3.7 |
| ✓ | ✓ | **94.2 ± 0.7** | **81.4 ± 0.8** | **86.3 ± 0.8** |

## 5  Conclusion

In this study, we present a novel SSL approach called $M^2$CRL, which integrates multi-view mask modeling with contrastive learning for endoscopic video analysis. Our method aims to address the lack of pixel-level information extraction in existing SSL methods for endoscopic videos. We leverage the unique characteristics of endoscopy to propose a frame-aggregated attention guided tube mask that captures pixel-level relationships across spatial-temporal dimensions for global views. Additionally, we utilize random tube mask to complement the details of local features from local views. Notably, the attention guidance is derived from multi-head self-attention maps extracted from a teacher model, without incurring additional computational costs. By integrating contrastive learning, our method not only maintains performance in dense prediction tasks but also ensures effectiveness in discriminative tasks. The experimental results on 10 publicly avaiable datasets demonstrate that our $M^2$CRL outperforms the state-of-the-art methods across multiple downstream vision tasks (*i.e.*, classification, segmentation, and detection).

## Acknowledgments

This work was supported in part by the National Natural Science Foundation of China under Grants 62272404 and 62372170, in part by the Natural Science Foundation of Hunan Province of China under Grants 2022JJ30571 and 2023JJ40638, and in part by the Research Foundation of Education Department of Hunan Province of China under Grant 23A0146.

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

# Supplementary Material

In this supplementary material, we first provide model implementation details for reproducibility in Section A. Next, we introduce additional ablation experiments in Section B. In Section C, we conduct another experiment on surgical phase recognition in downstream tasks. In Section D, we visualize more segmentation and detection results in downstream tasks and perform qualitative analysis. Subsequently, we provide the case study of frame-aggregated attention guided tube mask in Section E. Finally, we analyze the limitations and broader impacts of our work in Section F.

## A  Implementation Details

**Pre-training**  We employ data augmentation techniques such as random horizontal flipping, color jitter, gaussian blur, and exposure adjustment, and also utilize temporally consistent spatial enhancements [88] to all frames within a single view. The decoder is a lightweight one-layer head [16] and the model performs learning using a simple $\ell_1$ loss. The FLOPs of our model for a single execution are approximately 190G, with a parameter count of 121M. Pre-training are conducted on 4 Tesla A100 GPUs. The training parameters are shown in Table 7.

Table 7: **Pre-training settings.**

| config | value |
|---|---|
| optimizer | AdamW [89] |
| optimizer momentum | $\beta_1, \beta_2 = 0.9, 0.999$ |
| weight decay | $4e-2$ |
| base learning rate | $2e-5$ |
| learning rate schedule | cosine schedule [90] |
| warmup epochs | 10 |
| pretraining epochs | 30 |
| batch size | 12 |
| temperature parameters | $\tau_t, \tau_s = 0.04, 0.07$ |
| mask rate | $\rho = 0.9$ |
| attention areas threshold | $\gamma = 0.6$ |
| momentum coefficient | $\lambda = 0.996$ |

**Evaluation methodology**  For downstream fine-tuning, the procedure is as follows: (1) Classification: We use the PolypDiag [27] dataset, which includes 253 videos and 485,561 frames. Each video is annotated to indicate the presence or absence of a lesion. The dataset is divided into 71 normal videos without polyps and 102 abnormal videos with polyps for training, and 20 normal videos and 60 abnormal videos for testing. We sample 8 frames at a resolution of $224 \times 224$ from each video as input, utilizing a pre-trained model to initialize the backbone and appending randomly initialized linear layers for training 20 epochs. The SGD optimizer is employed, with the learning rate set to 1e-3, momentum to 0.9, and batch size to 4. (2) Segmentation: We use the CVC-12k [28] dataset, which includes 29 videos and 612 frames, with 20 videos allocated for training and 9 videos for testing. Each frame in the videos is annotated with ground truth masks (with a single class) to identify the regions covered by polyps. We employ TransUnet [91] as the segmentation decoder following the code of [91]. The AdamW optimizer is used to optimize the overall parameters by setting the learning rate as 1e-4, weight deacy as 5e-2 and the batchsize as 1. We resize the spatial size as $224 \times 224$ and fine-tune for 150 epochs. (3) Detection: We use the KUMC [29] dataset, which includes 53 videos and 19,832 frames. Each frame in each video is annotated with bounding boxes and polyp categories, with 36 videos allocated for training and 17 videos for testing. We employ STFT [92] to implement the detection task, fine-tuning for 24k iterations at a resolution of $640 \times 640$. The SGD optimizer is used to optimize the overall parameters by setting the learning rate as 2.5e-3, weight deacy as 1e-4 and momentum as 0.9. See the STFT [92] for more training details.

## B Additional Ablations

**Prediction target** From Table 8, it is observed that M$^2$CRL achieves better performance when using RGB pixel values as the reconstruction target. The model, which utilizes feature distillation as the prediction target, demonstrates decent results in classification tasks but shows a decline in performance in segmentation and detection tasks. This once again highlights the significant advantage of incorporating a reconstruction task involving masked patches for pixel-level tasks.

Table 8: **Prediction target**. The effect of pixel regression is better.

| prediction target | cla. | seg. | det. |
|---|---|---|---|
| feature distillation | 94.2 ± 0.4 | 77.9 ± 1.6 | 84.5 ± 1.0 |
| pixel regression | **94.2 ± 0.7** | **81.4 ± 0.8** | **86.3 ± 0.8** |

**Ablations on loss** The mask modeling component of our model follows SimMIM [16], which employs the $\ell_1$ loss function. The study demonstrates that different loss functions have minimal impact. To maintain consistency, we use the same loss function as SimMIM. Furthermore, we conduct ablation studies to demonstrate that different loss functions have a negligible effect on our results, as shown in Table 9.

Table 9: **Ablations on loss.**

| loss | cla. | seg. | det. |
|---|---|---|---|
| $\ell_1$ | 94.2 ± 0.7 | 81.4 ± 0.8 | 86.3 ± 0.8 |
| $\ell_2$ | 93.8 ± 0.7 | 82.0 ± 0.7 | 85.9 ± 1.5 |

**Ablation on different architectures** For a fair comparison, we use the weights for initialization as Endo-FM did. However, since this work does not provide weights for ViT variants, we are unable to conduct ablation experiments on different architectures with weight initialization. Consequently, we conduct a set of ablation experiments without weight initialization for the backbone, as shown in Table 10. We observe that the performance improvement is more pronounced with larger models due to their increased parameters and more complex structures, enabling them to capture more intricate features. In contrast, smaller models have limited feature extraction capabilities and cannot fully extract visual features. Although larger models exhibit stronger learning abilities, they are more prone to overfitting during training. Additionally, larger models require higher computational resources and longer training times. In conclusion, choosing ViT-B as the pre-trained backbone is a suitable compromise.

Table 10: **Ablation on different architectures.**

| backbone | cla. | seg. | det. |
|---|---|---|---|
| ViT-T/16 | 93.4 ± 0.9 | 76.8 ± 1.2 | 76.3 ± 2.4 |
| ViT-S/16 | 93.8 ± 0.4 | 78.2 ± 1.5 | 79.4 ± 0.7 |
| ViT-B/16 | 93.4 ± 0.9 | 80.5 ± 0.5 | 83.4 ± 2.8 |
| ViT-L/16 | 94.0 ± 0.9 | 83.2 ± 0.8 | 84.2 ± 2.0 |

Table 11: **Ablation on initialization status.**

| initialization status | cla. | seg. | det. |
|---|---|---|---|
| random | 93.4 ± 0.9 | 80.5 ± 0.5 | 83.4 ± 2.8 |
| kinetics weights | **94.2 ± 0.7** | **81.4 ± 0.8** | **86.3 ± 0.8** |

**Ablation on initialization status** We perform ablation experiments on the model initialization status. As shown in Table 11, the results indicate that M$^2$CRL without weight initialization performs slightly worse under the same pre-training conditions. This is because weight initialization accelerates model convergence and enhances model stability. However, for a fair comparison, we follow Endo-FM and use initialized weights.

## C  Surgical phase recognition

In the downstream tasks, we conduct surgical phase recognition experiments using the Cholec80 [93] dataset, which contains 80 complete laparoscopic cholecystectomy videos. This dataset is specifically designed to evaluate the performance of the model in automatic surgical phase recognition. To ensure a fair comparison with existing methods, we follow the data split and evaluation protocol described by [94], using 40 videos for training and 40 for testing. For evaluation, we adopt the relaxed boundary F1 score proposed by [95]. Since our backbone is based on Transformers, we do not use the CNN-based TCN [94] as a feature extractor during fine-tuning, so our result is based on single frames. As shown in Table 12, our method achieves superior performance. This experiment further verifies the robustness and effectiveness of our method across various endoscopic video tasks.

Table 12: **Surgical phase recognition.**

| Methods | F1 |
|---------|------|
| DINO | 77.6 |
| MoCo v2 | 81.7 |
| SimCLR | 84.5 |
| SwAV | 86.1 |
| Endo-FM | 87.5 |
| M$^2$CRL | **88.7** |

## D  Qualitative Evaluation

Fig. 3 illustrates the segmentation results of our method and other self-supervised pre-training methods on the CVC-12k dataset. Polyps are characterized by their blurry boundaries and significant shape variations, which undoubtedly add complexity to the segmentation task. Nevertheless, our method consistently outperforms other state-of-the-art self-supervised methods. Specifically, when polyps are very close and overlapping ($3^{rd}$ row of Fig. 3), our method produces smoother and more continuous edges, whereas other methods often result in blurred or fragmented edges. Additionally, some small polyps ($4^{th}$ row of Fig. 3) occupy very few pixels and are easily missed, our method successfully segments these small polyps without omission. Fig. 4 shows the detection results of our method and other self-supervised pre-training methods on the KUMC dataset. As shown in the figure, our method performs well in boundary identification and localization for small polyps ($1^{st}$ and $2^{nd}$ rows of Fig. 4). Furthermore, $3^{rd}$ and $4^{th}$ rows of Fig. 4 demonstrate that our method can also accurately identify and locate polyps even when they have low contrast and indistinct boundaries with surrounding tissues. Notably, compared to Endo-FM, a discriminative approach, our method significantly enhances the performance on pixel-level tasks.

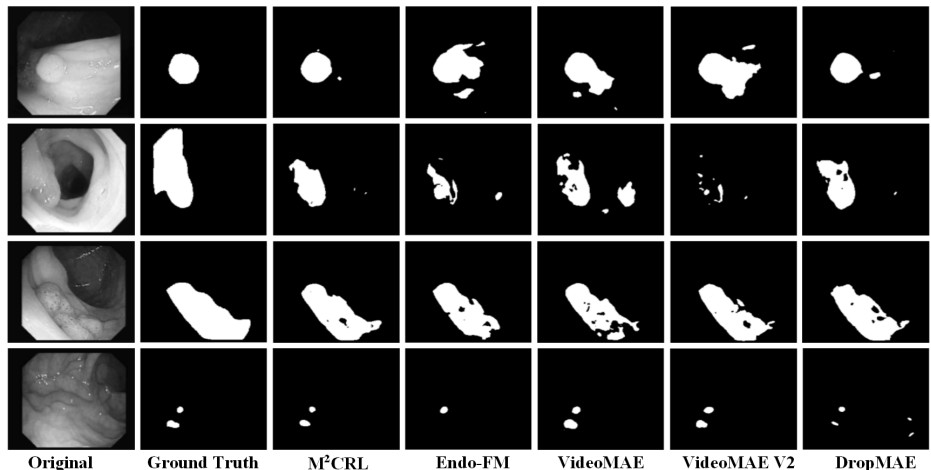

Figure 3: Qualitative results for segmentation task on the CVC-12k dataset.

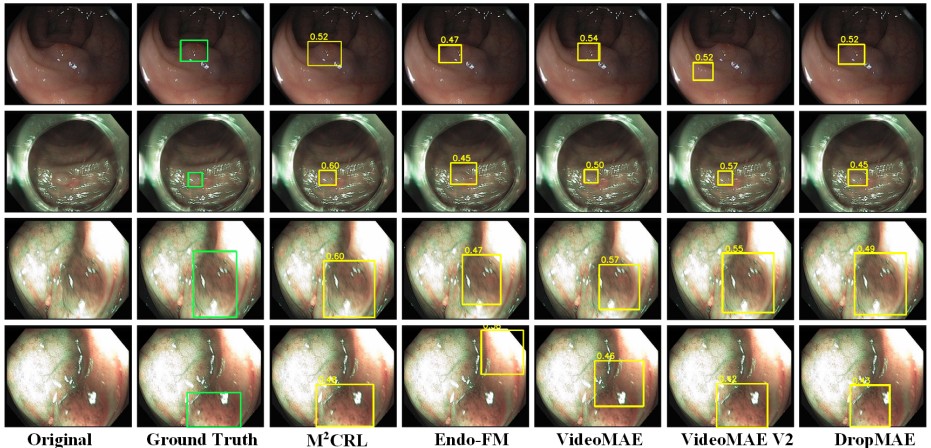

Figure 4: Qualitative results for detection task on the KUMC dataset.

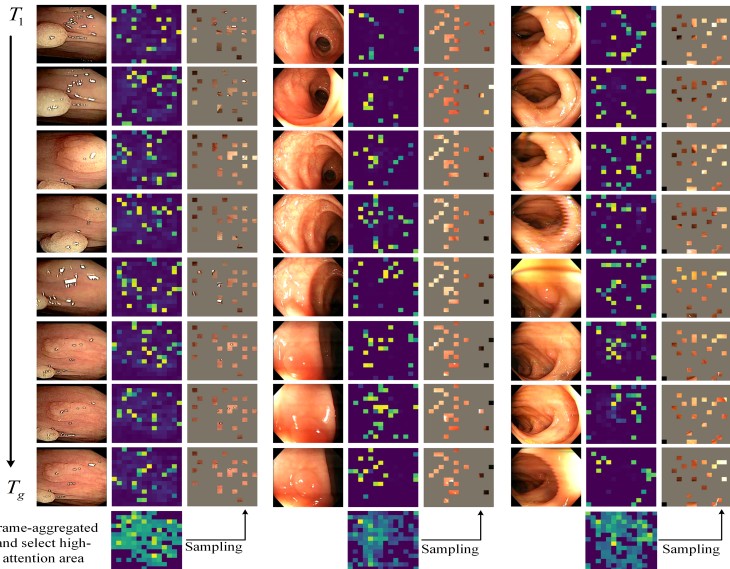

Figure 5: Illustration of our frame-aggregated attention guided tube masking strategy. We visualize spatial attention map with temporal information for each frame ($2^{nd}$ column), then aggregate attention maps for all frames and select area of high attention. We sample visible patches in this area ($3^{rd}$ column).

## E   Case Study

In Fig. 5, we visualize the attention maps of the ViT-B/16 model employing "the divided space-time attention" mechanism. We utilize the $[cls]$ token as a query and employ attention from the last transformer block. From the attention map of individual frames, we observe that the model does not distinctly learn the concept of endoscope object boundaries, attributed to the separate computations of attention in time and space. To enhance object concepts within endoscope video sequences, we aggregate attention across all frames of the video and then select a certain proportion of high-attention regions. This approach ensures that even in complex endoscopic video scenarios, key information within the video sequence is retained. Random masking at a high ratio in the video can obscure critical regions, thereby impeding the ability of the model to learn video representations. To address this issue, we select regions of high attention from the aggregated attention map for sampling visible tokens.

# F  Limitations and Broader Impacts

**Limitations**  Our work presents a multi-view masked contrastive representation learning ($M^2CRL$) framework for endoscopic video analysis. However, our current self-supervised pre-training method only utilizes RGB video streams and does not incorporate additional audio and text streams. In the future, we expect that audio and text data can provide more information for self-supervised pre-training. Furthermore, our study requires extensive pre-training, leading to significant energy consumption and reliance on high-performance computing hardware (GPU). These negative impacts underscore the necessity of considering environmental protection and resource conservation. In future work, we will adopt more efficient training methods and optimization strategies to address these issues.

**Broader impacts**  Our approach demonstrates the potential of SSL in endoscopic video analysis. By utilizing a large amount of unlabeled endoscopic video data for pre-training, we can reduce the dependence on costly annotated medical data, thus lowering healthcare expenses. Furthermore, our pre-trained models can be easily applied to various tasks such as classification, segmentation, and detection, thereby making valuable contributions to medical applications and enhancing the quality and efficiency of clinical disease diagnosis and healthcare services.

