# OpenReview forum: "Multi-view Masked Contrastive Representation Learning for Endoscopic Video Analysis"
_NeurIPS.cc/2024/Conference — NeurIPS 2024 poster_

### Official Review · Reviewer_u6Br · 2024-07-12

**Soundness:** 3
**Presentation:** 3
**Contribution:** 3
**Rating:** 6
**Confidence:** 4

**Summary:**

The authors propose a self-supervised learning regime for spatio-temporal data called multi-view masked contrastive learning which combines a frame-aggregated attention guided tube mask and a multi-view mask strategy using a student-teacher framework. The learnt representations are evaluated on multiple downstream tasks on publicly available datasets.

**Strengths:**

-	The manuscript is well and clearly written, different aspects are well motivated and explained.
-	The background section summarizes a lot of related literature and is easy to read. It gives a good overview of related work, particular challenges associated with endoscopy images and motivates the proposed approach.
-	The conducted experiments are exhaustive. The learnt representations are tested in multiple downstream tasks for their usefulness (classification, segmentation, detection) on three different publicly available datasets, comparing to a large number of recent competing methods. The results are averaged over three runs and show consistent improvements in all three tasks. The study includes an ablation study to report on the effect of different components of the proposed methods.
-	The method has been developed for application in endoscopy, but are relevant for spatio-temporal imaging in general and the type of pretraining is relevant for other medical imaging modalities with a temporal component.

**Weaknesses:**

-	Please provide some more information on the downstream tasks (how many classes, what kind of labels, how are they obtained,…)
-	Source code is not (yet) released.

Minor Remarks:

-	“temporal” is missing the “l” in a few places (e.g. l. 162,208)
-	“reach” -> “reaches” in l.281
-	I would suggest to remove the particular results from the abstract

**Questions:**

- How many classes do the datasets used for evaluations have in the case of classification and segmentation? What are they?
- Are any modifications needed to employ this method to other spatio-temporal data?

**Limitations:**

-	The potential negative societal impact is not sufficiently discussed. In the broad impact statement, the author discuss the need and benefits for their method one more time. I urge them to consider potential *negative* impact on society.

---

> ### Author Rebuttal · Authors · 2024-08-07
>
> We greatly appreciate your constructive and insightful comments. We address all weaknesses and questions below.
>
> **W1 and Q1. Detail information about downstream tasks.**
>
> Our downstream tasks include classification, segmentation, and detection, each addressing different aspects of endoscopic video analysis.
>
> * **Classification Task:** The classification task is a binary classification problem aimed at diagnosing polyps. We use the PolypDiag [1] dataset, which follows the gastroscope examination protocol and includes 253 videos and 485,561 frames. Each video is annotated to indicate the presence or absence of a lesion. The dataset is divided into 71 normal videos without polyps and 102 abnormal videos with polyps for training, and 20 normal videos and 60 abnormal videos for testing.
>
> * **Segmentation Task:** The segmentation task aims to perform polyp segmentation. We use the CVC-12k [2] dataset, which follows the colonoscopy examination protocol and serves as the official training dataset for the Automatic Polyp Detection sub-challenge of the MICCAI 2015 competition. This dataset includes 29 videos and 612 frames, with 20 videos allocated for training and 9 videos for testing. Each frame in the videos is annotated with ground truth masks (with a single class) to identify the regions covered by polyps.
>
> * **Detection Task:** The detection task aims to perform polyp detection. We use the KUMC [3] dataset, which follows the colonoscopy examination protocol and is sourced from the Kansas University Medical Center. This dataset includes 53 videos and 19,832 frames. Each frame in each video is annotated with bounding boxes and polyp categories, with 36 videos allocated for training and 17 videos for testing, including hyperplastic polyps and adenomatous polyps.
>
> For a fair comparison, all datasets are sourced from Endo-FM, and the data partitioning for all datasets follows Endo-FM guidelines. For more details on the experimental reproduction of downstream tasks, we will provide more detailed relevant hyperparameters in the appendix.
>
> **W2. Source code is not (yet) released.**
>
> We will release the code and pre-trained checkpoints upon acceptance. For reproducibility we build on the open source implementation of Endo-FM and provide all the relevant hyperparameters in the appendix.
>
> **W3. Minor Remarks.**
>
> Thank you for your valuable suggestions. We will remove the particular results from the Abstract and make careful writing revisions in subsequent revisions.
>
> **Q2. Are any modifications needed to employ this method to other spatio-temporal data?**
>
> Although our work is based on an endoscopic video dataset, it is important to note that the video format of endoscopic videos is the same as that of natural videos (Kinetics, SSv2). The basic techniques for processing and analyzing video data, such as frame extraction and video decoding, are essentially the same for both types. Therefore, applying our method to other spatio-temporal data does not necessitate specific modifications; rather, it only requires adherence to standard techniques for processing video data.
>
> **Limitations. The potential negative societal impact is not sufficiently discussed.**
>
> Our study involves self-supervised learning, which requires extensive pre-training leading to significant energy consumption. Additionally, large-scale model training requires high-performance computing hardware (GPUs), leading not only to increased hardware costs but also resource overconsumption and electronic waste. These negative impacts underscore the necessity of considering environmental protection and resource conservation. In future work, we will adopt more efficient training methods and optimization strategies to address these issues.
>
> [1] Tian Y, et al. Contrastive transformer-based multiple instance learning for weakly supervised polyp frame detection. MICCAI 2022.
>
> [2] Bernal J, et al. WM-DOVA maps for accurate polyp highlighting in colonoscopy. Computerized medical imaging and graphics 2015.
>
> [3] Li K, et al. Colonoscopy polyp detection and classification: Dataset creation and comparative evaluations. Plos one 2021.

---

> > ### Comment · Reviewer_u6Br · 2024-08-08
> > **Response to Rebuttal by Authors**
> >
> > I acknowledge having read the other reviewers' reviews as well as responses by the authors. I thank the authors for the detailed responses addressing everyone's questions and concerns. I will keep my original score.

---

> ### Author Response · Authors · 2024-08-09
> **Thanks for the feedback!**
>
> Thank you for recognizing our efforts in rebuttal. Your comments have greatly improved the quality and clarity of our paper. We appreciate your support!

---

### Official Review · Reviewer_BGWe · 2024-07-12

**Soundness:** 3
**Presentation:** 4
**Contribution:** 3
**Rating:** 7
**Confidence:** 4

**Summary:**

This work presents M$^2$CRL, a self-supervised learning method for representation learning of endoscopic videos. The method leverages a multi-view masking technique with attention-guided masking of global features and random spatiotemporal tube masking of local features. Both contrastive learning and masked autoencoding pretraining objectives are employed for stronger representation learning. The model is pretrained on 7 publicly available endoscopy video datasets and fine-tuned on 3 other datasets on a variety of tasks, outperforming other relevant baselines.

**Strengths:**

- The organization of the paper is excellent, with helpful use of bold text, logical flow from one passage to the next, and high-quality, information-dense illustrations and tables.
- The motivation is clearly laid out and the reference to prior related work is thorough.
- The experiments are very thorough and appear to be soundly conducted. Results demonstrate notable improvement upon existing competitive baselines, and ablation studies help showcase which elements of M$^2$CRL are most impactful.

**Weaknesses:**

- Experimental details could be clarified. Were all methods pretrained on the same union of 7 datasets as M$^2$CRL? Was Endo-FM used as is for downstream fine-tuning or retrained on this pretraining dataset? Section 4.2 could clarify these details. Additionally, how were hyperparameters selected and were they the same for each model and task? Appendix A should additionally include fine-tuning hyperparmeters.
- There are a few instances of confusing passages of writing. See examples in the section below.

**Questions:**

- Were all methods (perhaps except EndoFM) pretrained on the same union of 7 datasets as M$^2$CRL? Please clarify this in Section 4.2.
- It appears that the same hyperparameters were used for all methods and datasets – is this true? If so, how were hyperparameters selected? Please include full fine-tuning hyperparmaters in Appendix A alongside the pretraining hyperparameters.
- Several instances of awkward/confusing writing:
  - The passage from L225-233 is confusing, e.g.: “It is worth noting that the two students in Fig. 1(a) share weights, but are actually one student network. Both encoders share the same structure.” Further, perhaps explain why only local views are fed to the student network.
  - L267: “The results of other methods are taken from the comparative method of Endo-FM”. Is this saying these were the baselines considered in the EndoFM paper?

Minor comments:
- I would replace “mask strategy” with “masking strategy” throughout the paper.
- It might be useful to explain in words what Equation 5 represents.
- L275: I would refrain from saying “significant” without a statistical significance test.

**Limitations:**

Limitations are adequately addressed in Appendix E.

---

> ### Author Rebuttal · Authors · 2024-08-07
>
> We greatly appreciate your constructive and insightful comments. We address all weaknesses and questions below.
>
> **Q1. Were all methods pretrained on the same union of 7 datasets as M$^2$CRL?**
>
> Yes, for a fair comparison, all methods were pretrained on the same union of 7 datasets as our M$^2$CRL. We apologize for this and we will make careful revisions in subsequent versions.
>
> **Q2. Was Endo-FM used as is for downstream fine-tuning or retrained on this pretraining dataset?**
>
> Endo-FM [1] was originally pre-trained on this pretraining dataset, and its backbone was subsequently used for fine-tuning in downstream tasks. Endo-FM serves as our baseline, and the datasets we use are also provided and processed by it.
>
> **Q3. How were hyperparameters selected and were they the same for each model and task? Section 4.2 could clarify these details.**
>
> For the other methods, we used the hyperparameters as documented in their original papers or provided in their code. Similarly, for the hyperparameters of downstream tasks, we followed the guidelines outlined by Endo-FM [1], which are comprehensively explained in their paper and open-source code. We pretrained all methods on the same union of 7 datasets and then loaded them into the backbone of downstream tasks for fine-tuning. We apologize for this unclear presentation and we will make careful revisions in subsequent versions.
>
> **Q4. Appendix A should additionally include fine-tuning hyperparameters.**
>
> Thank you for your valuable suggestions. We will update Appendix A to include the fine-tuning hyperparameters as requested.
>
> **Q5. The passage from L225-233 is confusing.**
>
> We propose a Multi-view Masked Contrastive Representation Learning framework. In the manuscript, L225-233 primarily introduce the contrastive representation learning component of our framework.
>
> **L225-229** describe self-distillation and explain the relationship between contrastive learning and self-distillation. It is clarified that the contrastive representation learning component in our framework is achieved through self-distillation.
>
> **L229-233** provide a description of the specific workflow of our framework, including data generation, model composition, and data flow within the model. Our model employs a teacher-student architecture, consisting of a student network and a teacher network, which consists of a student network and a teacher network sharing the same structure. It should be noted that the two student networks depicted in Fig. 1(a) actually represent a single student network. We illustrated two student networks in the figure to more clearly convey the data flow.
>
> **Q6. Explain why only local views are fed to the student network.**
>
> We create two spatiotemporal views (global and local views) of the input endoscopic videos with varying spatial sizes and frame rates. The global views are fed into both the teacher and student networks, while the local views are only fed into the student network. Both the teacher and student networks process these views and predict one view from another in the latent feature space. This approach enables the model to learn spatiotemporal invariant features that can be transferred across different endoscopic domains and disease types.
>
> * **Cross-View Matching**: The global views are fed into the teacher network primarily because it provides stable and reliable outputs as references for the student network. These global views contain comprehensive information about the video, which assists the teacher network in generating accurate feature representations. The student network processes both the global and local views. By aligning the global features of the student network with those of the teacher network, the student network can learn rich global information. Furthermore, by aligning the local views of the student network with the global features of the teacher network, it can acquire more fine-grained feature representations. This cross-view matching approach involving global and local views facilitates a more comprehensive learning process for a rich feature representation within this model.
>
> * **Increased Learning Challenge**: Not feeding the local views to the teacher network introduces an information imbalance between the teacher and student networks. This design introduces a learning challenge for the model, as it requires the student network to exert more effort in extracting and integrating information from the local views to compensate for the lack of global information. Consequently, this approach enhances the contrastive learning process by necessitating that the student network maximize agreement between representations of different views despite their partial and incomplete nature. Ultimately, this design encourages the model to develop robust and discriminative features.
>
> **Q7. L267 Is this saying these were the baselines considered in the Endo-FM paper?**
>
> Endo-FM [1] compares several methods, including TimeSformer, CROP, FAME, ProViCo, VCL, and ST-Adapter. These methods are used for comparison rather than serving as baselines. The authors of Endo-FM pre-trained these methods on the same union of 7 datasets and then fine-tuned them on downstream tasks. Since we followed the same experimental procedure, we have directly utilized the recorded results from Endo-FM instead of re-running the experiments.
>
> **Q8. Minor comments:**
>
> Thank you for your comments. We will replace the term “mask strategy” with “masking strategy” throughout the entire manuscript in subsequent versions. We will also provide a detailed textual explanation of what Equation (5) represents. Furthermore, we will remove “significant” from Line 275. We sincerely appreciate once again your valuable suggestions, which have greatly contributed to the improvement of our manuscript.
>
> [1] Wang Z, et al. Foundation Model for Endoscopy Video Analysis via Large-scale Self-supervised Pre-train. MICCAI 2023.

---

> > ### Comment · Reviewer_BGWe · 2024-08-07
> >
> > I acknowledge that I have read the authors' rebuttal and thank them for the clarifications. I will maintain my score.

---

> ### Author Response · Authors · 2024-08-09
> **Thanks for the feedback!**
>
> We sincerely appreciate your positive feedback on our manuscript. We're glad our efforts to address your questions were satisfactory. Thank you for your support.

---

### Official Review · Reviewer_R5Z9 · 2024-07-13

**Soundness:** 3
**Presentation:** 2
**Contribution:** 2
**Rating:** 5
**Confidence:** 3

**Summary:**

The paper proposes a representation learning framework that combines masked pretraining strategies with contrastive learning methods. Particularly, this framework aims to generate a representation learning approach that can work with downstream tasks requiring dense pixel-level representations (for image segmentation) or discriminative features (for image classification). The masking strategy is guided by the aggregation of the attention layers of a teacher model over different frames of the endoscopic video, while the contrastive learning framework follows a self-distillation approach. Testing is performed on colonoscopic datasets and on three downstream tasks, including classification, detection, and segmentation.

**Strengths:**

* Representation learning in the medical domain can have an impact on the development of models for medical image analysis.

* The model is validated against different works and on three downstream tasks.

* Ablation experiments show the contribution of the components.

**Weaknesses:**

* Downstream tasks are evaluated on a single endoscopic modality

* Similarly, a cross-validation like approach that keep the downstream tasks but change the testing dataset, could help to support the results.

**Questions:**

Most of the datasets, including the downstream tasks, are related to colonoscopy. Given the intention to develop a method to operate with endoscopy in general, it would be interesting to see the performance in downstream tasks that involve other endoscopic modalities.

It appears that the combination of masking and contrastive learning is more beneficial for the classification and detection tasks, while the improvements for the segmentation tasks are still at a similar level as previous works. Would it be possible to elaborate on why the segmentation tasks might not take significant benefits compared with the other tasks?

**Limitations:**

Limitations are discussed in the supplementary material of the paper.

---

> ### Author Rebuttal · Authors · 2024-08-07
>
> We greatly appreciate your constructive and insightful comments. We address all weaknesses and questions below.
>
> **W1 and Q1. Downstream tasks are evaluated on a single endoscopic modality.**
>
> In our study, we used multiple publicly available endoscopic video datasets. These datasets cover 3 types of endoscopic procedures (colonoscopy, gastroscopy, and laparoscopy) and 10+ different diseases. We believe that this comprehensive and large-scale dataset is valuable for endoscopic research (as detailed in Table 1). In downstream tasks, classification is for gastroscopy, while segmentation and detection are for colonoscopy.
>
> Based on your suggestion, we have added the Cholec80 dataset in downstream tasks. This dataset contains 80 complete laparoscopic cholecystectomy videos and is specifically designed for laparoscopic surgical recognition. Followed by previous works [1, 2], we used 40, 8, and 32 videos from Cholec80 as our training, validation, and test sets, respectively. We compared our approach with the classical endoscopic foundation model Endo-FM [3] and the masked video modeling method VideoMAE [4] using F1-score in Table 2. The experimental results demonstrate that our approach achieves superior performance. This experiment further validates the robustness and effectiveness of our approach across diverse endoscopic video tasks.
>
> Table 1. Details of pre-train and downstream datasets
>
> | **Phase**  |  **Dataset**   |  **Provider**  | **Videos** |      **Protocol**       |
> | :--------: | :------------: | :------------: | :--------: | :---------------------: |
> | Pre-train  |  Colonoscopic  |      CNRS      |    210     |       colonoscope       |
> |            |    SUN-SEG     |      ANU       |    1018    |       colonoscope       |
> |            |  LDPolypVideo  |      USTC      |    237     |       colonoscope       |
> |            |  Hyper-Kvasir  |     Simula     |    5704    |       gastroscope       |
> |            | Kvasir-Capsule |     Simula     |    1000    |       gastroscope       |
> |            | CholecTriplet  |     BIDMC      |    580     |       laparoscope       |
> |            | Renji-Hospital | Renji Hospital | 16494/7653 | colonoscope/gastroscope |
> | Downstream |   PolypDiag    |    Adelaide    |    253     |       gastroscope       |
> |            |    CVC-12k     |      UAB       |     29     |       colonoscope       |
> |            |      KUMC      |     Kansas     |     53     |       colonoscope       |
> |            |    Cholec80    |     CAMMA      |     80     |       laparoscope       |
>
> Table 2. Results of surgical phase recognition
>
> | **Methods**  |  **Recog.**  |
> | :----------: | :----------: |
> | Endo-FM [3]  |   82.2±0.8   |
> | VideoMAE [4] |   73.7±1.4   |
> |    M$^2$CRL     | **85.0±0.4** |
>
> **W2. A cross-validation like approach.**
>
> We conducted five-fold cross-validation on the 3 downstream tasks, and the experimental results are presented in Table 3. We observe that the results of cross-validation are lower than those recorded in our original manuscript. We think that this difference is caused by the different ways of data division. For a fair comparison, we strictly followed the public partitioning of the datasets for conducting experiments in our study. Considering that there is no existing work to cross-validate this dataset, we also cross-validated the classical endoscopy foundation model Endo-FM [3] for comparison. The results show that our method outperforms Endo-FM in all three tasks.
>
> Table 3. Results of cross validation for 3 downstream tasks
>
> | **Methods** | **Cla.** | **Seg.** | **Det.** |
> | :---------: | :------: | :------: | :------: |
> |   Endo-FM   | 84.5±4.5 | 55.5±7.0 | 80.5±4.2 |
> |    M$^2$CRL    | 87.5±4.9 | 58.6±8.5 | 82.0±4.2 |
>
> **Q2. Why the segmentation tasks might not take significant benefits compared with the other tasks?**
>
> Due to varying requirements in different downstream tasks, the features acquired through different pre-training methods may exhibit distinct performances on specific downstream tasks. Previous studies, such as VideoMAE, DropMAE, and VideoMAE V2, have shown impressive performance in segmentation tasks due to their reliance on masked modeling. The strength of masked modeling lies in capturing rich pixel-level information, making these single-task pretraining approaches particularly effective for dense pixel-level tasks like segmentation. However, their performance tends to underperform in structural tasks that focus on global features and the holistic understanding of objects in images or videos.
>
> Our method integrates masked modeling with contrastive learning, which not only encourages the model to capture fine-grained pixel-level features but also compels it to learn comprehensive discriminative representations. During the pre-training process, the model needs to carefully consider feature selection tradeoffs across multiple pre-tasks. This consideration may result in some downstream tasks exhibiting less prominent performance compared to single pre-task training. Although the improvement of our method in segmentation tasks is not as significant as that using mask modeling techniques alone, our method still achieves a significant improvement in segmentation tasks compared to the method using only contrast learning. Overall, our method achieves robust performance across both structural and pixel-level tasks, highlighting its versatility and effectiveness in learning both detailed and global features.
>
> [1] Ramesh S, et al. Dissecting self-supervised learning methods for surgical computer vision. Medical Image Analysis, 2023.
>
> [2] Czempiel T, et al. Surgical phase recognition with multi-stage temporal convolutional networks. MICCAI 2020.
>
> [3] Wang Z, et al. Foundation Model for Endoscopy Video Analysis via Large-scale Self-supervised Pre-train. MICCAI 2023.
>
> [4] Tong Z, et al. VideoMAE: Masked Autoencoders are Data-Efficient Learners for Self-Supervised Video Pre-Training. NeurIPS 2022.

---

> > ### Comment · Reviewer_R5Z9 · 2024-08-13
> > **Renspose to rebuttal**
> >
> > I thank the authors for the response. I will update my score to accept.
> > Thanks.

---

> ### Author Response · Authors · 2024-08-14
> **Thanks for the feedback!**
>
> Thank you for recognizing our efforts in the rebuttal. We sincerely appreciate your consideration in raising our paper's score to 7: Accept. Your feedback has been invaluable to our work. Thank you for your support.

---

### Official Review · Reviewer_wGJR · 2024-07-14

**Soundness:** 3
**Presentation:** 3
**Contribution:** 2
**Rating:** 4
**Confidence:** 5

**Summary:**

The paper proposes a pre-training approach called Multi-view Masked Contrastive Representation Learning for endoscopic videos. The approach combines self-distillation and masked video modeling under multi-view setting. To consider the characteristics of inter-frame instability and small inter-class differences of endoscopic videos, the paper introduces a frame aggregated attention guided tube masking strategy to capture global spatio-temporal representation and employs random tube masking on local views to capture local representations. The approach is pre-trained on seven endoscopic datasets and fine-tuned on three additional datasets. Experiments show that it outperforms the baselines on classification, segmentation and detection tasks.

**Strengths:**

- The paper is easy to read.
- The paper shows the combination of self-distillation and masked video modeling for pre-training ViT-B model using endoscopic videos.
- Frame-aggregated attention guided tube masking (FAGTM) to learn global spatio-temporal representation learning.
- Experiments on multiple tasks to show the efficacy of the approach.

**Weaknesses:**

- Although the pre-training approach is proposed for endoscopic videos, the novelty is limited. It's a combination of self-distillation and masked video modeling.
- FAGTM is also an extension of either [1] and [2] which propose attention guide masking strategy for image based pre-training. The paper merely aggregates the attention for all frames and use the mean of that to guide the masking.
- The paper mentions in section 3.3, "This self-distillation method of self-supervision is also considered a form of contrastive learning". Can the authors please give remark on why self-distillation is a form of contrastive learning?
- Which block of the teacher ViT-B is used for FAGTM? Ablation study would be great.
- Did the author pre-train ViT-B from the scratch? or initialized from some weights?
- Ablation on number of epochs during pre-training is missing
- Only ViT-B is used in the experiment? Different architecture should be studied too.
- Did the authors also pre-train VideoMAE and other baselines using your dataset? Can the authors show some results using kinetics pre-trained SSL weights?
- More baselines should be compared with. For example, MME[3], AdaMAE[4], and other masked video modeling approach.
- What did the author use as an evaluation or metric to stop pre-training?
- Can FAGTM be used on local views?
- The approach looks very sensitive to $\gamma$. Table 3 shows the impact of it on all the tasks.
- Is the masking ratio used for FAGTM and random tube masking same? Ablation study on the impact of different masking ratio for each of the masking strategy would be useful.
- The paper only uses linear layer for the reconstruction objective. Study of different decoders would be helpful. In masked video modeling, most of the approaches used asymmetric encoder and decoder design. It would be great to pre-train baselines like VideoMAE with linear layer decoder for a better comparison.
- Most of the recent approaches use L2 loss for the reconstruction objective. Comparison of L1 and L2 loss is missing and can the authors please give a remark on why L1 loss is preferred?
- Given the approach is mostly empirical with limited novelty, the performance on other datasets like Cholec80, a surgical phase recognition benchmark dataset, would be great.

- There are some typos in the paper: section 3.2.1 'spatiotempora' -> temporal, table2 'gloabl' -> global

[1] What to Hide from Your Students: Attention-Guided Masked Image Modeling, ECCV 2022

[2] Good helper is around you: Attention-driven Masked Image Modeling, AAAI 2023.

[3] Masked Motion Encoding for Self-Supervised Video Representation Learning, CVPR 2023

[4] AdaMAE: Adaptive Masking for Efficient Spatiotemporal Learning with Masked Autoencoders, CVPR 2023

**Questions:**

Please see the weakness section for the questions and suggestions.

**Limitations:**

Yes, the authors have adequately addressed the limitations.

---

> ### Author Rebuttal · Authors · 2024-08-07
>
> We greatly appreciate your constructive and insightful comments. We address all weaknesses and questions below.
>
> **W1. The novelty.**
>
> Existing self-supervised pre-training methods for endoscopic videos predominantly rely on contrastive learning. However, using contrast learning alone is not sufficient to capture the fine-grained feature representations required for endoscopic videos.
>
> * To address the limitations of current methods, we have integrated contrastive learning with masked modeling to effectively acquire endoscopic video representations that possess both comprehensive discriminative capability and fine-grained perceptive ability.
>
> * Unlike the general approach of combining self-distillation and masking modeling, our method specifically designed a novel multi-view masking strategy for endoscopic video features, which significantly improved the model performance. This strategy involves utilizing the frame-aggregated attention guided tube mask to capture global-level spatiotemporal contextual relationships from the global views, while also employing the random tube mask to focus on local variations from the local views.
>
> * We conducted extensive experiments on 10 endoscopic video datasets to evaluate the performance of M$^2$CRL in comparison to other methods. Experimental results demonstrate the superiority of our method.
>
> This work holds inherent value in clinical practice. The development of robust pretrained models for endoscopic video analysis through self-supervised learning can effectively support various downstream tasks, ultimately enhancing clinical workflow efficiency.
>
> **W2. Innovation in FAGTM.**
>
> Unlike [1] and [2], our work focuses on video rather than static images. Videos have a specific temporal dimension that should be considered in the mask design process to fully leverage spatiotemporal information. Furthermore, endoscopic videos present unique challenges due to their domain-specific characteristics. In this study, we propose a multi-view masking strategy tailored to endoscopic video features, which enhances the model’s robustness and effectiveness.
>
> Specifically, to address the instability of inter-frame variations in endoscopic videos, we designed the FAGTM to capture global spatiotemporal representations. Traditional image-based attention mechanisms only focus on the spatial information of the current frame, neglecting dependencies across the entire video sequence. In contrast, our FAGTM aggregates the attention of frames to the mean value to obtain a simplified and holistic attention distribution. This attention distribution is capable of obtaining an approximation of the critical region of the video that are being attended to from both the temporal and spatial dimensions, while reducing the impact of excessive variations in individual frames or regions. Thus, our FAGTM provides more comprehensive and context-aware guidance for the masking process.
>
> **W9. Comparison with more methods.**
>
> As you suggested, we have included comparisons with MME, AdaMAE, and UTM methods, as shown in Table 5. The results indicate that our method achieves the best performance in classification and detection tasks and performs moderately well in the segmentation task. This is primarily due to the fact that these mask modeling methods excel in capturing rich pixel-level information, making them particularly effective for dense pixel-level tasks such as segmentation. However, their performance tends to underperform in structural tasks that focus on global features and the holistic understanding of objects in images or videos. Our method integrates masked modeling with contrastive learning, which not only encourages the model to capture fine-grained pixel-level features but also compels it to learn comprehensive discriminative representations. This is friendly for both pixel-level and structured tasks, but it also increases the complexity of model pre-training. During the pre-training process, the model needs to carefully consider feature selection tradeoffs across multiple pre-tasks. This consideration may result in some downstream tasks exhibiting less prominent performance compared to single pre-task training. Overall, our method achieves robust performance across both structural and pixel-level tasks, highlighting its versatility and effectiveness in learning both detailed and global features.
>
> Table 5. Comparison with more methods
>
> | **Methods** |   **Cla.**   |   **Seg.**   |   **Det.**   |
> | :---------: | :----------: | :----------: | :----------: |
> |  MME [10]   |   92.3±1.0   |   81.9±0.7   |   84.8±0.9   |
> | AdaMAE [11] |   92.0±0.3   |   **82.3±0.4**   |   83.4±1.6   |
> |  UTM [12]   |   93.2±0.3   |   80.8±0.5   |   84.0±1.1   |
> |  M$^2$CRL   | **94.2±0.7** | 81.4±0.8 | **86.3±0.8** |
>
> **W16. The performance on other datasets like Cholec80.**
>
> Based on your suggestion, we have added the Cholec80 dataset in our downstream tasks. We compared our approach with the classical endoscopic foundation model Endo-FM [4] and the masked video modeling method VideoMAE [5] using F1-score in Table 9. The experimental results demonstrate that our approach achieves superior performance.
>
> Table 9. Results of surgical phase recognition
>
> | **Methods**  |  **Recog.**  |
> | :----------: | :----------: |
> | Endo-FM [4]  |   82.2±0.8   |
> | VideoMAE [5] |   73.7±1.4   |
> |   M$^2$CRL   | **85.0±0.4** |
>
> **Additional comment:** We would have liked to include the rest of answers to the questions mentioned by Reviewer wGJR (marked as W3, W4, etc..). Unfortunately, we did not have enough space in this rebuttal box. As soon as the discussion phase will begin, we will include the mentioned answers in an additional comment for the reviewer.

---

> ### Author Response · Authors · 2024-08-07
> **Response to additional questions (Part 1/3)**
>
> As we mentioned in the main rebuttal, we include the answers to the additional questions of Reviewer wGJR. We hope that this helps to address all remaining concerns and we thank again for taking the time to review our work.
>
> **W3. Why self-distillation is a form of contrastive learning？**
>
> The pretraining tasks for self-supervised learning are summarized into two categories: contrastive and generative. Contrastive methods focus on maximizing the similarity between different views of the same image after augmentation, while also potentially minimizing the similarity between views of different images after augmentation. Self-distillation is optimized by comparing the feature representations of the teacher and student models to extract a closer representation from the same image. In our work, the self-distillation component also serves to align features across different views, which follows the paradigm of contrastive learning. Therefore, self-distillation can be considered a form of contrastive learning. Furthermore, several other studies [1-3] have also classified self-distillation in self-supervised learning as a type of contrastive learning.
>
> **W4. Ablation on teacher’s block used for FAGTM.**
>
> In our study, we used the last layer block of the teacher ViT-B for FAGTM. The higher layer block incorporates lower-level features and object-level semantic information, offering comprehensive and abstract features that are essential for the model. Thus, it effectively guides the student network in masking. Table 1 shows that it is most beneficial for FAGTM to utilize the last layer block of the teacher network.
>
> Table 1. Ablations on blocks
>
> | **Blocks** |   **Cla.**   |   **Seg.**   |   **Det.**   |
> | :--------: | :----------: | :----------: | :----------: |
> |     4      |   91.8±0.7   |   76.6±1.5   |   83.6±1.1   |
> |     8      |   92.5±0.4   |   79.7±2.2   |   84.8±1.0   |
> |     10     |   93.9±1.0   |   80.6±0.9   |   85.9±1.7   |
> |     12     | **94.2±0.7** | **81.4±0.8** | **86.3±0.8** |
>
> **W5. Whether to initialize ViT-B?**
>
> We pre-trained ViT-B with weight initialization to accelerate convergence and enhance training stability. This way also ensures consistency with the baseline (Endo-FM), which similarly employed weight initialization for pre-training ViT-B.
>
> **W6 and W10. Ablation on pre-training epochs. Evaluation or metric to stop pre-training.**
>
> For a fair comparison, we used the same number of epochs as Endo-FM. As you suggested, we performed ablation studies on different pre-training epochs, as shown in Table 2. While increasing the number of epochs generally enhances performance, excessive training may result in diminishing returns or overfitting, particularly when dealing with a smaller endoscopic dataset compared to Kinetics. This overfitting can impair the model’s generalization.
>
> During the pre-training process, we stopped pre-training based on monitoring the loss on the training set. If there is no significant decrease in training loss over multiple epochs or if the loss curve becomes flat, it suggests that the model has likely acquired most of the features. At this point, we consider stopping pre-training.
>
> Table 2. Ablations on epochs
>
> | **Epochs** |   **Cla.**   |   **Seg.**   |   **Det.**   |
> | :--------: | :----------: | :----------: | :----------: |
> |     10     |   84.6±3.1   |   73.8±1.7   |   83.6±1.3   |
> |     20     |   92.7±0.4   |   78.2±1.5   |   85.8±2.5   |
> |     30     | **94.2±0.7** | **81.4±0.8** | **86.3±0.8** |
> |     40     |   93.7±0.6   |   81.0±0.4   |   86.0±1.5   |
> |     50     |   94.7±0.9   |   80.9±0.6   |   85.5±1.0   |
>
> **W7. Ablation on different architecture.**
>
> For a fair comparison, we used the weights for initialization as Endo-FM did. However, since this work did not provide weights for ViT variants, we were unable to conduct ablation experiments on different architectures with weight initialization. Consequently, we conducted a set of ablation experiments without weight initialization for the backbone, as shown in Table 3. It was observed that the performance improvement is more pronounced with larger models due to their increased parameters and more complex structures, enabling them to capture more intricate features. In contrast, smaller models have limited feature extraction capabilities and cannot fully extract visual features. Although larger models exhibit stronger learning abilities, they are more prone to overfitting during training. Additionally, larger models require higher computational resources and longer training times. In conclusion, choosing ViT-B as the pre-trained backbone is a suitable compromise.
>
> Table 3. Ablations on architecture
>
> | **Backbone** | **Cla.** | **Seg.** | **Det.** |
> | :----------: | :------: | :------: | :------: |
> |   ViT-T/16   | 93.4±0.9 | 76.8±1.2 | 76.3±2.4 |
> |   ViT-S/16   | 93.8±0.4 | 78.2±1.5 | 79.4±0.7 |
> |   ViT-B/16   | 93.4±0.9 | 80.5±0.5 | 83.4±2.8 |
> |   ViT-L/16   | 94.0±0.9 | 83.2±0.8 | 84.2±2.0 |

---

> ### Author Response · Authors · 2024-08-07
> **Response to additional questions (Part 2/3)**
>
> **W8. Pre-train VideoMAE and other baselines using your dataset? Some results using kinetics pre-trained SSL weights.**
>
> Yes, to ensure the fairness of experiments, all compared methods were pretrained on the same union of 7 datasets as our M$^2$CRL. As you suggested, Table 4 presented results using SSL weights pretrained on Kinetics for 3 downstream tasks. We observed that the performance of these methods pretrained on Kinetics is comparable to those pretrained on the endoscopic datasets. This can be attributed to the significantly larger size of the Kinetics dataset compared to the endoscopic datasets. Pretraining models on such a large dataset allows them to learn more generalized features and patterns. Consequently, when these models are transferred to endoscopic downstream tasks, their robust feature extraction capabilities allow them to effectively fine-tune on the endoscopic datasets, thereby adapting efficiently to the new tasks. Moreover, despite the substantial differences in content between endoscopic datasets and Kinetics, both consist of color images, which means they share some similar fundamental visual features such as edges and colors. Therefore, models pretrained on Kinetics can capture these common features to some extent, enabling them to perform well when transferred to endoscopic downstream tasks.
>
> Table 4. Results of using Kinetics pre-trained SSL weights on 3 downstream tasks
>
> | **Methods**  | **Cla.** | **Seg.** | **Det.** |
> | :----------: | :------: | :------: | :------: |
> |   SVT [7]    | 88.7±0.7 | 74.8±1.1 | 84.8±0.9 |
> | VideoMAE [5] | 90.9±0.6 | 81.1±0.3 | 85.5±0.9 |
> | DropMAE [9]  | 85.8±0.9 | 81.2±0.4 | 83.2±0.3 |
>
> **W11. Can FAGTM be used on local views?**
>
> FAGTM cannot be used on local views. In our method, the global views are fed into both the teacher and student networks, allowing the teacher to generate attention corresponding to the global views to guide the student model in masking. However, the local views are only fed into the student networks, thus the teacher does not have attention for the local view to guide masking. This approach is inspired by self-supervised visual transformers [6, 7], where a teacher-student framework is used for contrastive pre-training. In this framework, different views of the video are processed by the teacher and student networks, and predictions are made between views in the latent feature space, enabling the model to learn spatiotemporally invariant features.
>
> **W12. The approach looks very sensitive to $\gamma$.**
>
> The parameter $\gamma$ serves as the threshold for FAGTM, which is designed to allow the model to sample visible tokens from its attention regions and mask the rest in a reasonable manner. This ensures that our method can effectively perform reconstruction tasks even at a high masking rate. A lower value of $\gamma$ implies selecting visible patches from a smaller high-attention region, potentially leading to an overemphasis on non-critical areas during reconstruction, thus contradicting the objectives of the self-supervised pretraining task. Conversely, a higher value of $\gamma$ means selecting visible patches from a broader region, which may dilute the focus on high-attention areas and adversely affect the model’s learning efficiency.
>
> **W14. Ablation of VideoMAE using linear layer decoder.**
>
> As you suggested, we evaluated VideoMAE on downstream tasks after pre-training with a linear layer decoder. As shown in Table 7, the impact of the decoder on the experimental results is minimal and almost negligible. The prediction head can be of arbitrary form and capacity, as long as its input conforms with the encoder output and its output accomplishes the prediction target. This has already been validated in SimMIM [8].
>
> Table 7. Ablations on decoder
>
> | **Decoder**  | **Cla.** | **Seg.** | **Det.** |
> | :----------: | :------: | :------: | :------: |
> | Linear layer | 91.2±0.8 | 81.2±0.3 | 82.6±1.4 |
> |  Asymmetric  | 91.4±0.8 | 80.9±1.0 | 82.8±1.9 |
>
> **W15. Why use L1 loss for reconstruction?**
>
> The mask modeling component of our model follows SimMIM [8], which employs the L1 loss function. The study demonstrates that different loss functions have minimal impact. To maintain consistency, we used the same loss function as SimMIM. Furthermore, we conducted ablation studies to demonstrate that different loss functions have a negligible effect on our results.
>
> Table 8. Ablations on loss
>
> | **Loss** | **Cla.** | **Seg.** | **Det.** |
> | :------: | :------: | :------: | :------: |
> |    L1    | 94.2±0.7 | 81.4±0.8 | 86.3±0.8 |
> |    L2    | 93.8±0.7 | 82.0±0.7 | 85.9±1.5 |
>
>
>
> **W17. Some typos.**
>
> We apologize for any typos. We will proofread the whole manuscript carefully and make revisions in subsequent versions.

---

> ### Author Response · Authors · 2024-08-07
> **Response to additional questions (Part 3/3)**
>
> **W13. Ablation on different masking ratio for each of the masking strategy.**
>
> We conducted ablation experiments on different masking rates for FAGTM (global views) and RTM (local views). As our work is video-related, we aligned our masking rates with those used in other video masking modeling studies. As shown in Table 6, both excessively high and low masking rates are unfavorable for endoscopic representation learning. A high masking rate increases the difficulty of pre-training and hinders the ability of the model to learn effective representations, while a low masking rate reduces the challenge for the model and fails to fully utilize the advantages of masked learning to extract potential features from the video.
>
> Table 6. Masking ratio
>
> | **FAGTM (Global)** | **RTM (Local)** | **Cla.** | **Seg.** | **Det.** |
> | :----------------: | :-------------: | :------: | :------: | :------: |
> |        95%         |       95%       | 94.0±0.3 | 81.3±0.4 | 85.1±1.1 |
> |                    |       90%       | 93.8±0.9 | 80.7±0.7 | 86.2±1.3 |
> |                    |       85%       | 93.2±0.7 | 78.5±0.6 | 84.9±2.3 |
> |                    |       75%       | 92.6±0.4 | 77.4±1.7 | 85.2±2.1 |
> |        90%         |       95%       | 93.8±1.4 | 80.5±0.7 | 85.8±0.9 |
> |                    |       90%       | 94.2±0.7 | 81.4±0.8 | 86.3±0.8 |
> |                    |       85%       | 93.8±0.8 | 81.4±1.7 | 85.6±2.2 |
> |                    |       75%       | 93.2±0.9 | 78.5±1.9 | 84.8±1.3 |
> |        85%         |       95%       | 93.2±0.8 | 79.9±0.4 | 83.1±1.5 |
> |                    |       90%       | 93.8±0.2 | 81.2±0.2 | 83.8±0.9 |
> |                    |       85%       | 94.0±0.4 | 80.5±1.0 | 85.1±1.8 |
> |                    |       75%       | 92.5±1.2 | 79.6±0.7 | 83.8±2.5 |
> |        75%         |       95%       | 91.7±1.3 | 76.8±1.8 | 84.2±2.1 |
> |                    |       90%       | 91.3±0.3 | 79.0±2.2 | 84.0±1.3 |
> |                    |       85%       | 91.8±0.2 | 77.5±1.9 | 83.8±0.4 |
> |                    |       75%       | 91.2±0.7 | 74.6±1.4 | 85.0±0.8 |
>
>
>
> [1] Dong X, et al. Maskclip: Masked self-distillation advances contrastive language-image pretraining. CVPR 2023.
>
> [2] Gupta A, et al. Siamese masked autoencoders. NeurIPS 2023.
>
> [3] Chen X, et al. Context autoencoder for self-supervised representation learning. IJCV 2024.
>
> [4] Wang Z, et al. Foundation Model for Endoscopy Video Analysis via Large-scale Self-supervised Pre-train. MICCAI 2023.
>
> [5] Tong Z, et al. VideoMAE: Masked Autoencoders are Data-Efficient Learners for Self-Supervised Video Pre-Training. NeurIPS 2022.
>
> [6] Caron M, et al. Emerging properties in self-supervised vision transformers. ICCV 2021.
>
> [7] Ranasinghe K, et al. Self-supervised video transformer. CVPR 2022.
>
> [8] Xie Z, et al. SimMIM: A simple framework for masked image modeling. CVPR 2022.
>
> [9] Wu Q, et al. Dropmae: Masked autoencoders with spatial-attention dropout for tracking tasks. CVPR 2023.
>
> [10] Sun X, et al. Masked motion encoding for self-supervised video representation learning. CVPR 2023.
>
> [11] Bandara, et al. Adamae: Adaptive masking for efficient spatiotemporal learning with masked autoencoders. CVPR 2023.
>
> [12] Li K, et al. Unmasked teacher: Towards training-efficient video foundation models. ICCV 2023.

---

> ### Comment · Reviewer_wGJR · 2024-08-13
> **Response to the authors (Part 1/3)**
>
> I sincerely thanks authors for clarifications and experiments. However, I still have some questions regarding novelty and experiments. I would also suggest the authors to add all new experiments in the revision.
>
> $\textbf{Novelty}$
>
> I agree that for surgical videos, most of the papers have been using contrastive learning for pre-training, but recently, there are a few papers that have studied self-distillation and contrastive leanring. EndoFM uses global and local views and apply self-distillation. [1] comprehensively studies the self-distillation and contrastive learning approaches for endoscopic videos. The only component that differentiates this approach with other methods is applying masked modeling on both global and local views. FAGTM module is mere extension of attention guided masking for images.
>
> [1] Dissecting Self-Supervised Learning Methods for Surgical Computer Vision. Medical Image Analysis 2023
>
> $\textbf{Comparison with more methods.}$
>
> Thanks for adding more experiments. Did you pre-train MME and AdaMAE from the scratch or did you fine-tune these models for endoscopic downstream task?
>
> $\textbf{Cholec80}$
>
> I see some missing baselines like TeCNO[2], LoViT[3]
>
> [2] Tecno: Surgical phase recognition with multi-stage temporal convolutional networks
>
> [3] LoViT: Long Video Transformer for Surgical Phase Recognition
>
> In the above paper [1], table 6 shows F1 score on Cholec-80 dataset. SimCLR, when fine-tuned with 10 labeled videos, can achieve 85.0 and can go upto 93.6 when fine-tuned will all labeled videos. Same goes for other self-supervised approaches like DINO, MoCo v2 and SwaV with temporal model TCN.

---

> > ### Author Response · Authors · 2024-08-14
> > **Re: Response to the authors (Part 1/3)**
> >
> > Thank you for your thoughtful feedback on our submission. These valuable suggestions have improved the clarity and quality of our work. We have further addressed all comments and questions below.
> >
> > **Novelty**
> >
> > We summarized our novelty as follows:
> >
> > * **Why did we apply masked modeling?** As pointed out by the reviewer, existing self-supervised pre-training methods for endoscopic videos primarily rely on contrastive learning. Contrastive learning naturally endows the pre-trained model with strong instance discrimination capabilities. However, relying solely on contrastive learning is insufficient to capture the fine-grained feature representations required for endoscopic videos. Therefore, we combine contrastive learning with masked modeling to acquire endoscopic video representations that possess both comprehensive discriminative capability and fine-grained perceptive ability, effectively addressing the limitations of contrastive learning in capturing dense pixel dependencies. ***This is our first major contribution.***
> > * **What is unique about our masking strategy?** We have developed a novel multi-view masking strategy specifically tailored to the characteristics of endoscopic video data to address two key challenges inherent in endoscopic videos. First, the instability of inter-frame variations is a prominent issue in endoscopic videos. These variations are caused by factors such as camera movement, instrument manipulation, and the uneven distribution of the lesion area. However, traditional attention-guided masking strategies fail to adequately address the global spatiotemporal nature of videos, and therefore cannot effectively consider the instability between frames in endoscopic videos. To address this, we designed the FAGTM strategy from global views, which aggregates features across multiple frames to capture comprehensive spatiotemporal information. Second, endoscopic videos often exhibit minimal inter-class differences, where lesion characteristics closely resemble those of the surrounding normal tissue. This requires the model to capture fine-grained pixel-level details to distinguish these subtle differences. Thus, we employed a random tube masking strategy from local views to learn finer local details. The effectiveness of our multi-view masking strategy is demonstrated through ablation experiments. ***This is our second major contribution.***
> > * **Clinical significance.**  We conducted extensive experiments on 10 endoscopic video datasets to evaluate the performance of M$^2$CRL in comparison to other methods. The experimental results demonstrate the superiority of our method, which holds inherent value in clinical practice. The development of robust pretrained models for endoscopic video analysis through self-supervised learning can effectively support various downstream tasks, ultimately enhancing clinical workflow efficiency.
> >
> >
> >
> > **Comparison with more methods**
> >
> > We pre-trained MME and AdaMAE from scratch before fine-tuning them for the endoscopic downstream task.
> >
> >
> >
> > **Cholec80**
> >
> > Regarding the supplementary experiments in Table 6 of Paper [1]: The differences in experimental results arise from the use of different data partitioning methods. The results you mentioned in Table 6 are supplementary experiments for comparison with external methods, following the approach in reference [2], which uses 40 videos as the training set and 40 videos as the test set. In contrast, our method follows the data partitioning method in Table 4 of Paper [1], using 40 videos as the training set, 8 videos as the validation set, and 32 videos as the test set. As shown in the table below, our method achieved optimal performance under this setup. Furthermore, we will include additional experimental results based on the data partitioning outlined in Table 6 in the revised manuscript.
> >
> > Regarding the baselines of TeCNO and LoViT: Our data partitioning and evaluation metrics followed the methods outlined in Paper [1]. However, due to different evaluation metrics (we use F1 score, while TeCNO uses accuracy), direct comparison with TeCNO is not feasible. Additionally, LoViT utilizes different data partitioning and evaluation metrics from ours, which also prevents direct comparison. We appreciate your reminder. In future research, we will conduct additional experiments, including adding TeCNO and LoViT methods to the comparisons presented in Table 6 of Paper [1], to provide more comprehensive results.
> >
> > Table 9. Result of surgical phase recognition.
> >
> > | **Methods** |  **F1**  |
> > | :---------: | :------: |
> > |    DINO     |   81.6   |
> > |   MoCo v2   |   79.6   |
> > |   SimCLR    |   81.1   |
> > |    SwAV     |   79.5   |
> > |   Endo-FM   | 82.2±0.8 |
> > |  VideoMAE   | 73.7±1.4 |
> > |  M$^2$CRL   | 85.0±0.4 |
> >
> > [1].Dissecting Self-Supervised Learning Methods for Surgical Computer Vision. MedIA 2023.
> >
> > [2].Semi-supervised learning with progressive unlabeled data excavation for label-efficient surgical workflow recognition.  MedIA 2023.

---

> ### Comment · Reviewer_wGJR · 2024-08-13
> **Response to Authors (Part 2/3)**
>
> $\textbf{initialize ViT-B?}$
>
> What would be the performance gap if ViT-B is not initialized with weights? Can we pre-train M$^{2}$CRL from the scratch?
>
> $\textbf{Pre-training epochs?}$
>
> The pre-training epochs is too low considering the we are pre-training a transformer using some proxy tasks. I think merely looking at the loss doesn't really justify to stop the pre-training.
>
> $\textbf{Ablation on different architecture?}$
>
> I think it would be great to pre-train a larger model using M$^{2}$CRL and compare it random initialization or imagenet/kinetics weight. The current set of experiments don't really tell the whole story. The only thing I can infer is the performance of M$^{2}$CRL can be achieved without weight initialization or pre-training.
>
> $\textbf{VideoMAE and other baselines?}$
>
> Did you pre-train them from initialized SSL weights? From the Table 4, these SSL methods achieve the same performance without pre-training on pixel-level tasks like segmentation and detection. M$^{2}$CRL only achieves significant performance on classification task.
>
> $\textbf{Approach sensitive to $\gamma$}$
>
> There's a drastic difference in performance when using 0.5 and 0.6 values. I understand the performance variability is using smaller or higher value, but performance gap in 0.5 and 0.6 don't correlate with the authors justifications.

---

> > ### Comment · Reviewer_wGJR · 2024-08-13
> > **Response to Authors (Part 3/3)**
> >
> > $\textbf{Ablation on masking ratio}$
> >
> > It looks like with 0.95 making ratio for both global and local views yield comparable performance which doesn't correlate when the authors say both excessively high and low masking rates are unfavorable for endoscopic representation learning.
> >
> > Given all my above mentioned concerns regarding novelty and experiments, I will maintain my original score of 4.

---

> > > ### Author Response · Authors · 2024-08-14
> > > **Re: Response to Authors (Part 3/3)**
> > >
> > > **Ablation on masking ratio**
> > >
> > > We agree your point that a masking rate of 0.95 can also yield comparable performance results. Our statement that excessively high and low masking rates are unfavorable for endoscopic representation learning is a relative observation. Compared to the optimal masking rate, a higher masking rate increases the difficulty of pre-training, hindering the model’s ability to learn effective representations. Conversely, a lower masking rate reduces the challenge for the model, preventing it from fully capitalizing on the benefits of masked reconstruction for extracting latent features from the video.
> > >
> > > Although our ablation experiments have shown that a masking rate of 0.95 can achieve comparable performance, its effectiveness in three downstream tasks is lower than that of a 0.9 masking rate. This suggests that at a masking rate of 0.95, the model is placed in a relatively unfavorable learning situation, resulting in suboptimal results.
> > >
> > >
> > >
> > > **Overall**.
> > > We have re-summarized the innovation of our methods with a special emphasis on our research motivation and method design. At the same time, for the ablation experiments, we have addressed each comment point-to-point. If our response has resolved your concerns on our paper, we will greatly appreciate it if you could re-evaluate our paper. We are also willing and ready to engage in discussions, if you have any further questions.

---

> > ### Author Response · Authors · 2024-08-14
> > **Re: Response to Authors (Part 2/3) (1)**
> >
> > **Initialize ViT-B?**
> >
> > We can pre-train M$^2$CRL from scratch. As shown in the ablation study Table 10, the results indicate that M$^2$CRL without weight initialization performs slightly worse under the same pre-training conditions. This is because weight initialization accelerates model convergence and enhances model stability. However, for a fair comparison, we followed Endo-FM and used initialized weights.
> >
> > Table 10. Initialization status.
> >
> > | **Initialization status** |  **Cla.**  |  **Seg.**  |  **Det.**  |
> > | :-----------------------: | :--------: | :--------: | :--------: |
> > |          Random           | 93.4 ± 0.9 | 80.5 ± 0.5 | 83.4 ± 2.8 |
> > |     Kinetics weights      | 94.2 ± 0.7 | 81.4 ± 0.8 | 86.3 ± 0.8 |
> >
> >
> >
> > **Pre-training epoch?**
> >
> > We acknowledge your concerns regarding the pre-training epochs. we aligned with Endo-FM by using initialized weights and the same number of epochs when pre-training M$^2$CRL. Our model achieved excellent performance under these settings, thus demonstrating the comparability and effectiveness of our method.
> > During the pre-training process, if the training loss does not significantly decrease over multiple epochs or if the loss curve flattens, it suggests that the model may have been adequately trained. We select checkpoints from different epochs based on the convergence of the loss and determine the appropriate number of pre-training epochs by balancing computational time and performance.
> >
> >
> >
> > **Alation on different architecture?**
> >
> > We agree with your perspective that pre-training a larger M$^2$CRL model using random initialization or Kinetics weights can achieve great performance. Larger models have more parameters, which enhance their learning capacity, allowing them to capture more complex patterns and subtle differences in the data. To ensure the fairness of our experimental comparisons, we followed the Endo-FM by using weight initialization and achieved superior performance under the same training epochs, which also demonstrates the effectiveness of our method.
> > However, since Endo-FM does not provide ViT variant weights, we were unable to conduct ablation experiments using Kinetics weights for different architectures. Consequently, we conducted a set of ablation experiments without weight initialization for the backbone, as shown in Table 3. Additionally, in Table 10, we compared our method with random initialization and weight loading. The results indicate that M$^2$CRL with weight loading performs better under the same pre-training conditions, as weight loading can accelerate model convergence and enhance model stability.
> >
> >
> >
> > **VideoMAE and other baseline?**
> >
> > We did not use initialized SSL weights for pre-training; but refer to the original setup of the comparison methods.
> >
> > The results from Table 4 demonstrate that single masked modeling pre-training methods, such as VideoMAE and DropMAE, exhibit strong performance in fine-tuning pixel-level tasks for endoscopic downstream tasks following pre-training on the Kinetics dataset. This can be attributed to the extensive size of the Kinetics dataset, which enables the model to acquire a more diverse range of features and patterns through pre-training. Furthermore, both endoscopic data and Kinetics data are comprised of color images with shared fundamental visual characteristics. As a result, the weights pre-trained on Kinetics can be effectively transferred to endoscopic tasks.
> >
> > The advantage of masked modeling lies in its ability to capture rich pixel-level information, making it particularly effective for tasks requiring dense pixel-level processing. In contrast, the SVT method in Table 4, which is based on contrastive learning, does not perform as well as masked modeling methods in pixel-level tasks. Our method combines contrastive learning and masked modeling, enabling the model to capture both fine-grained pixel-level features and comprehensive discriminative features. This hybrid pre-training strategy may result in some downstream tasks not performing as prominently as single pre-training tasks.
> >
> > Compared to Table 4, our method demonstrates a significant improvement in classification tasks. This is due to the fact that classification is a relatively straightforward structured task. Our model was pre-trained on an endoscopic dataset, which enabled it to effectively learn the features and patterns of endoscopic images. However, segmentation and detection tasks, being more complex in nature, may not exhibit as substantial an improvement under the same training conditions.
> >
> > Overall, our method achieves robust performance in both structured and pixel-level tasks, demonstrating its versatility and effectiveness in learning both detailed and global features.

---

> > ### Author Response · Authors · 2024-08-14
> > **Re: Response to Authors (Part 2/3) (2)**
> >
> > **Approach sensitive to $\gamma$**
> >
> > The parameter $\gamma$ is the threshold for FAGTM, used to guide the model in sampling visible tokens from high-attention regions. Due to the feature of endoscopic videos, where the camera moves within the body and inter-frame variations are unstable, different regions of the video frames exhibit significant changes. When $\gamma$ = 0.5, the model considers half of the area as high-attention regions and samples visible tokens from them. In this case, the model samples within a more concentrated attention area, which can lead to insufficient capture of the extensive variations in endoscopic videos during reconstruction.
> > Conversely, selecting a relatively larger high-attention region to sample visible tokens enables the model to better adapt to the significant inter-frame variations in endoscopic videos and facilitates the capture of important content across frames. However, it is important to note that the $\gamma$ value should not be too large, as a higher $\gamma$ value implies sampling visible tokens from a wider area, which could potentially dilute the focus on high-attention regions and consequently impact the model’s learning efficiency. Through ablation experiments, we ultimately chose $\gamma$ = 0.6.

---

### Author Rebuttal · Authors · 2024-08-07

Dear reviewers and AC,



We sincerely appreciate your valuable time and effort spent reviewing our manuscript. We thank all reviewers for their useful comments, positive consideration and relevant feedback on our paper. It seems that the reviews are positive in general and acknowledges our main contributions and soundness of our work. Our submission has received four ratings, including one accept (7), one weak accept (6), one borderline accept (5), and one borderline reject (4).



The valuable feedback from the reviewers has significantly contributed to enhancing the quality of our manuscript. We have supplemented all ablation experiments according to the reviewers’ comments and suggestions. We have addressed each comment and question individually below and we would be glad to engage in discussion in case of more questions or concerns exist. We will also update the main paper in the future with the main requested changes for improvement. Furthermore, we kindly request Reviewer wGJR to reconsider our work after reviewing our response. Your reconsideration will be highly valued.



Based on the comments from the reviewers, we have summarized the strengths of our paper as follows:

* **Motivation: [Reviewer BGWe, u6Br]** The motivation is clearly laid out and the reference to prior related work is thorough. Different aspects are well motivated and explained. It gives a good overview of related work, particular challenges associated with endoscopy images and motivates the proposed approach.

* **Method: [Reviewer wGJR, R5Z9, BGWe, u6Br]** The paper shows the combination of self-distillation and masked video modeling for pre-training ViT-B model using endoscopic videos. Frame-aggregated attention guided tube mask (FAGTM) to learn global spatio-temporal representation learning. Ablation experiments show the contribution of the components. Ablation studies help showcase which elements of M$^2$CRL are most impactful. The study includes an ablation study to report on the effect of different components of the proposed methods.

* **Experiment: [Reviewer wGJR, R5Z9, BGWe, u6Br]** Experiments on multiple tasks to show the efficacy of the approach. The model is validated against different works and on three downstream tasks. The experiments are very thorough and soundly conducted. The conducted experiments are exhaustive. The results are averaged over three runs and show consistent improvements in all three tasks.

* **Expression: [Reviewer wGJR, BGWe, u6Br]** The paper is easy to read. The organization of the paper is excellent, with helpful use of bold text, logical flow from one passage to the next, and high-quality, information-dense illustrations and tables. The manuscript is well and clearly written. The background section summarizes a lot of related literature.

* **Impact: [Reviewer R5Z9, BGWe, u6Br]** Representation learning in the medical domain can have an impact on the development of models for medical image analysis. Results demonstrate notable improvement upon existing competitive baselines. The method has been developed for application in endoscopy, but are relevant for spatio-temporal imaging in general and the type of pretraining is relevant for other medical imaging modalities with a temporal component.



Here we also present the response to two questions below.

_**Concern about novelty of the model**_

We have summarized our novelty as follows:

* Existing self-supervised pre-training methods for endoscopic videos predominantly rely on contrastive learning. However, using contrastive learning alone is not sufficient to capture the fine-grained feature representations required for endoscopic videos. To address the limitations of current methods, we have integrated contrastive learning with masked modeling to effectively acquire endoscopic video representations that possess both comprehensive discriminative capability and fine-grained perceptive ability.

* Given the characteristics of inter-frame instability and small inter-class differences in endoscopic videos, we propose a multi-view mask strategy. Specifically, we introduce a frame-aggregated attention guided tube mask strategy for the global views, which aggregates features from multiple frames to capture global spatiotemporal information. Simultaneously, a random tube mask strategy is employed from the local views, enabling the model to focus on local features.

* Extensive experiments have verified that our M$^2$CRL significantly enhances the quality of endoscopic video representation learning and exhibits excellent generalization capabilities in multiple downstream tasks.

_**Concern about experiment to other datasets or endoscopy modalities**_

* **[Reviewer wGJR]:** The performance on other datasets like Cholec80, a surgical phase recognition benchmark dataset, would be great.

- **[Reviewer R5Z9]:** It would be interesting to see the performance in downstream tasks that involve other endoscopic modalities.

In our study, we used multiple publicly available endoscopic video datasets, provided by research groups worldwide and previous EndoVis challenges. These datasets cover 3 types of endoscopic procedures (colonoscopy, gastroscopy, and laparoscopy) and 10+ different diseases. We believe that this comprehensive and large-scale dataset is valuable for endoscopic research. In downstream tasks, classification is for gastroscopy, while segmentation and detection are for colonoscopy. Based on reviewer’s suggestion, we have added the Cholec80 dataset in downstream tasks. The results demonstrate that our approach achieves superior performance. This experiment further validates the robustness and effectiveness of our approach across diverse endoscopic video tasks.


We strongly believe that M$^2$CRL can be a useful addition to the NeurIPS community, in particular, due to the enhanced manuscript by reviewers’ comments helping us better deliver the effectiveness of our method.


Thank you very much!

Best regards,

Authors

---

### Decision · Program_Chairs · 2024-09-25

**Decision:**

Accept (poster)

**Comment:**

The authors proposed a pre-training scheme for endoscopic video data by adapting contrastive learning and masked modeling together. A novel masking strategy via frame-aggregated attention is also proposed to enhance the global spatiotemporal contextual relationships. The manuscript is well-written overall, with clarified technical novelty (after the rebuttal and discussion) and solid experimental results on ten public datasets (superior performance compared to prior SSL methods). In the rebuttal, the authors thoroughly addressed the reviewers' questions and supplemented the relevant results. Overall, the paper contributes novel insights and advancements to the pre-training task in medical imaging.